# Gradient-reading and mechano-effector machinery for netrin-1-induced axon guidance

Kentarou Baba[1], Wataru Yoshida[1†], Michinori Toriyama[1†], Tadayuki Shimada[1‡], Colleen F Manning[2], Michiko Saito[1§], Kenji Kohno[1], James S Trimmer[2], Rikiya Watanabe[3], Naoyuki Inagaki[1*]

[1]Division of Biological Science, Nara Institute of Science and Technology, Ikoma, Japan; [2]Department of Neurobiology, Physiology and Behavior, University of California, Davis, Davis, United States; [3]Department of Applied Chemistry, Graduate School of Engineering, University of Tokyo, Tokyo, Japan

**\*For correspondence:**
ninagaki@bs.naist.jp

[†]These authors contributed equally to this work

**Present address:** [‡]Synaptic Plasticity Project, Tokyo Metropolitan Institute of Medical Science, Setagaya-ku, Japan; [§]Bio-science Research Center, Kyoto Pharmaceutical University, Kyoto, Japan

**Competing interests:** The authors declare that no competing interests exist.

**Abstract** Growth cones navigate axonal projection in response to guidance cues. However, it is unclear how they can decide the migratory direction by transducing the local spatial cues into protrusive forces. Here we show that knockout mice of *Shootin1* display abnormal projection of the forebrain commissural axons, a phenotype similar to that of the axon guidance molecule netrin-1. Shallow gradients of netrin-1 elicited highly polarized Pak1-mediated phosphorylation of shootin1 within growth cones. We demonstrate that netrin-1–elicited shootin1 phosphorylation increases shootin1 interaction with the cell adhesion molecule L1-CAM; this, in turn, promotes F-actin–adhesion coupling and concomitant generation of forces for growth cone migration. Moreover, the spatially regulated shootin1 phosphorylation within growth cones is required for axon turning induced by netrin-1 gradients. Our study defines a mechano-effector for netrin-1 signaling and demonstrates that shootin1 phosphorylation is a critical readout for netrin-1 gradients that results in a directional mechanoresponse for axon guidance.
DOI: https://doi.org/10.7554/eLife.34593.001

## Introduction

Axon guidance is a critical step for the formation and regeneration of neuronal networks. More than a century ago Ramón y Cajal identified the growth cone at the tip of extending axons, and proposed that it senses extracellular chemical cues and produces force for axon guidance (*Cajal, 1890*; *Sotelo, 2002*; *Vitriol and Zheng, 2012*). Accumulating evidence indicates that growth cones are indeed guided by extracellular molecules (*Huber et al., 2003*; *Lowery and Van Vactor, 2009*; *Kolodkin and Tessier-Lavigne, 2011*) and generate traction forces (*Chan and Odde, 2008*; *Koch et al., 2012*). Furthermore, analyses with microfluidic devices have shown that growth cones can navigate in response to extremely shallow gradients of diffusible and substrate-bound chemical cues in the microenvironment (*Baier and Bonhoeffer, 1992*; *Rosoff et al., 2004*; *Xiao et al., 2014*).

Netrin-1 is one of the best-characterized axon guidance molecules (*Ishii et al., 1992*; *Serafini et al., 1994*; *Lai Wing Sun et al., 2011*). Extracellular gradients of netrin-1 elicit growth cone attraction in vitro (*Kennedy et al., 1994*; *Serafini et al., 1994*; *Hong et al., 1999*; *Bhattacharjee et al., 2010*; *Fothergill et al., 2014*). Mice lacking netrin-1 or its receptor deleted in colorectal cancer (DCC) show impaired projection and guidance of axons in the ventral spinal commissure and forebrain commissures (*Serafini et al., 1996*; *Fazeli et al., 1997*; *Bin et al., 2015*; *Yung et al., 2015*). The intracellular signaling pathways involved in netrin-1–induced axonal chemoattraction have been extensively analyzed. For example, stimulation of DCC by netrin-1 activates

**eLife digest** Neurons communicate with each other by forming intricate webs that link cells together according to a precise pattern. A neuron can connect to another by growing a branch-like structure known as the axon. To contact the correct neuron, the axon must develop and thread its way to exactly the right place in the brain. Scientists know that the tip of the axon is extraordinarily sensitive to gradients of certain molecules in its surroundings, which guide the budding structure towards its final destination.

In particular, two molecules seem to play an important part in this process: netrin-1, which is a protein found outside cells that attracts a growing axon, and shootin1a, which is present inside neurons. Previous studies have shown that netrin-1 can trigger a cascade of reactions that activates shootin1a. In turn, activated shootin1a molecules join the internal skeleton of the cell with L1-CAM, a molecule that attaches the neuron to its surroundings. If the internal skeleton is the engine of the axon, L1-CAMs are the wheels, and shootin1a the clutch. However, it is not clear whether shootin1a is involved in guiding growing axons, and how it could help neurons 'understand' and react to gradients of netrin-1.

Here, Baba et al. discover that when shootin1a is absent in mice, the axons do not develop properly. Further experiments in rat neurons show that if there is a little more netrin-1 on one side of the tip of an axon, this switches on the shootin1a molecules on that edge. Activated shootin1a promote interactions between the internal skeleton and L1-CAM, helping the axon curve towards the area that has more netrin-1. In fact, if the activated shootin1a is present everywhere on the axon, and not just on one side, the structure can develop, but not turn. Taken together, the results suggest that shootin1a can read the gradients of netrin-1 and then coordinate the turning of a growing axon in response.

Wound healing, immune responses or formation of organs are just a few examples of processes that rely on cells moving in an orderly manner through the body. Dissecting how axons are guided through their development may shed light on the migration of cells in general. Ultimately, this could help scientists to understand disorders such as birth abnormalities or neurological disabilities, which arise when this process goes awry.

DOI: https://doi.org/10.7554/eLife.34593.002

Cdc42 and Rac1 and their downstream kinase Pak1, thereby inducing growth cone expansion and axon extension (*Li et al., 2002*; *Shekarabi and Kennedy, 2002*; *Shekarabi et al., 2005*; *Briançon-Marjollet et al., 2008*; *Demarco et al., 2012*). The actin regulatory proteins ENA/VASP and N-WASP are also required for netrin-1–induced growth cone expansion (*Lebrand et al., 2004*; *Shekarabi et al., 2005*). In addition, a number of signaling molecules, including phospholipase Cγ, $Ca^{2+}$, cAMP, phosphatidylinositol-3 kinase (PI3K), ERK1/2, focal adhesion kinase (FAK) and Src, are reported to be involved in netrin-1–induced axonal chemoattraction (*Song and Poo, 2001*; *Lowery and Van Vactor, 2009*; *Lai Wing Sun et al., 2011*; *Moore et al., 2012*; *Gomez and Letourneau, 2014*; *Sutherland et al., 2014*). However, despite the significant progress in identifying the signaling pathways, little is known about how the netrin-1 signal, as a spatial cue, is converted into the directional force required for axon guidance. Moreover, a molecular understanding of how shallow gradients of chemical cues are read out to guide axons is lacking (*Quinn and Wadsworth, 2008*; *Hegemann and Peter, 2017*).

Shootin1, recently renamed shootin1a (*Higashiguchi et al., 2016*), is a brain-specific protein involved in axon outgrowth (*Toriyama et al., 2006*; *Sapir et al., 2013*). At the leading edge of growth cones, actin filaments (F-actins) polymerize and disassemble proximally, thereby undergoing retrograde flow (*Forscher and Smith, 1988*; *Katoh et al., 1999*). Shootin1a interacts with F-actin retrograde flow via cortactin (*Weed and Parsons, 2001*) and couples the F-actin flow with extracellular adhesive substrates (*Shimada et al., 2008*; *Kubo et al., 2015*) through the cell adhesion molecule L1-CAM (*Rathjen and Schachner, 1984*; *Kamiguchi et al., 1998*). We previously reported that Pak1 (*Manser et al., 1994*), upon activation by netrin-1, phosphorylates shootin1a (*Toriyama et al., 2013*); this in turn enhances shootin1a interaction with F-actin flow, thereby producing traction force on the substrate (*Toriyama et al., 2013*; *Kubo et al., 2015*). However, whether shootin1a mediates

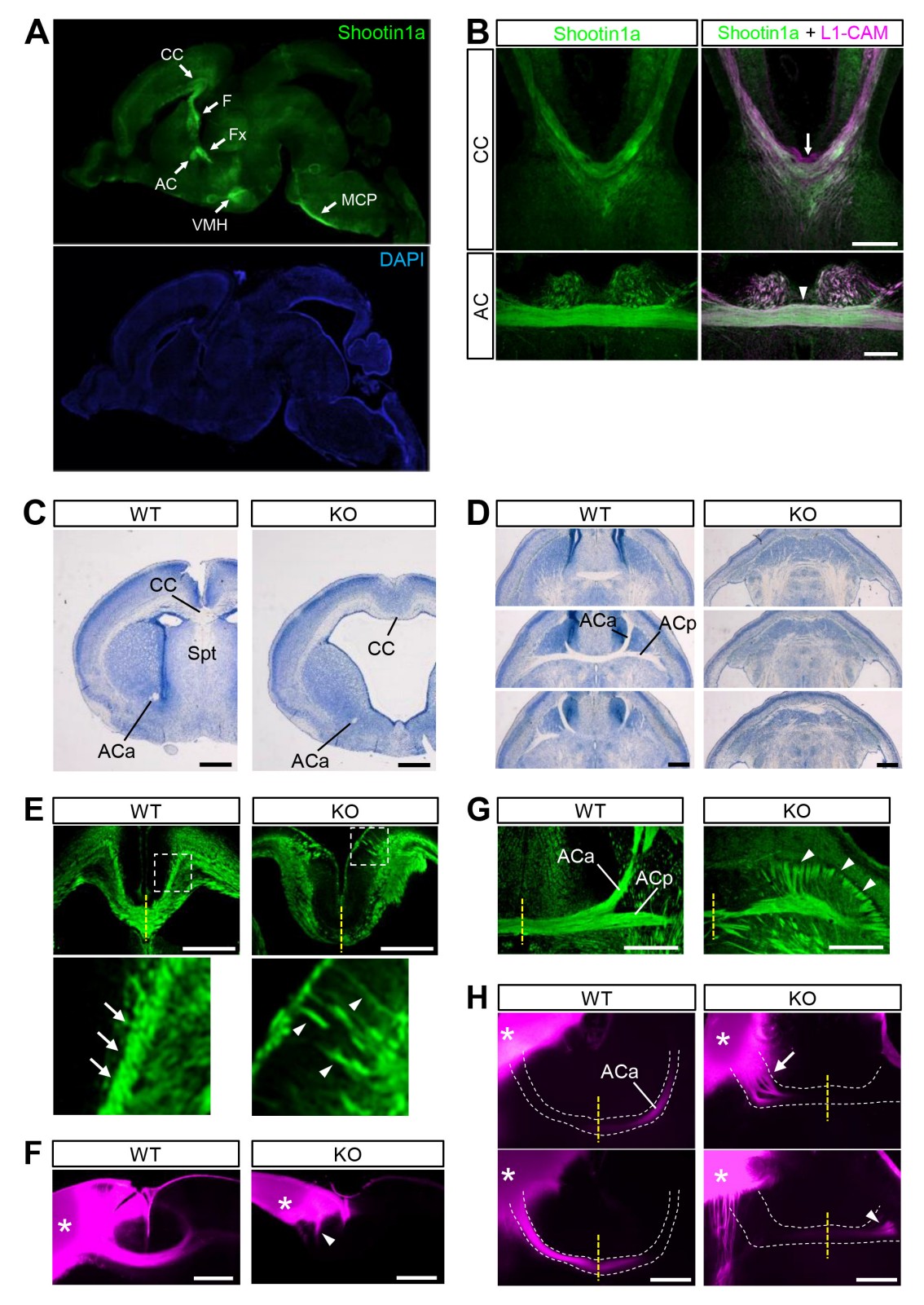

**Figure 1.** *Shootin1* knockout mice display abnormal projection of forebrain commissural axons. (A) A representative sagittal section of a P0 mouse brain immunolabeled with shootin1a antibody (green) and counterstained with DAPI (blue). (B) Coronal sections of E16.5 mouse brains double-immunolabeled with anti-shootin1a (green) and anti-L1-CAM (magenta) antibodies. The arrow and arrowhead indicate the corpus callosum and anterior commissure, respectively. (C) Coronal sections of the forebrain of wild-type and *Shootin1* knockout mice at P0 stained for Nissl substance. (D) Serial

*Figure 1 continued on next page*

*Figure 1 continued*

horizontal sections of the ventral forebrain of wild-type and *Shootin1* knockout mice at P0 stained for Nissl substance. (**E**) Coronal sections of wild-type and *Shootin1* knockout mouse brains at P0 immunolabeled with anti-L1-CAM antibody (green). Ectopic axonal projections were observed in the neocortex (arrowheads). In the knockout mice, the prominent axonal tracts observed in the intermediate zone of the neocortex of wild-type mice (arrows) were undetectable and ectopic axonal projections were observed (arrowheads). Lower panels show enlarged views of the rectangles. (**F**) Coronal sections of wild-type and *Shootin1* knockout mouse brains at P0. DiI crystals (magenta) were placed into the neocortex (asterisks) to label callosal axons. An arrowhead indicates incomplete contralateral projections of callosal axons. (**G**) Horizontal sections of wild-type and *Shootin1* knockout mouse brains at P0 immunolabeled with anti-L1-CAM antibody (green). In *Shootin1* knockout mice, the bundling of the commissural axons was disrupted (arrowheads). (**H**) Horizontal sections of wild-type and *Shootin1* knockout mouse brains at P0. DiI crystals (magenta) were placed in the anterior piriform cortex (asterisks) to label the anterior limb of the anterior commissure. Defasciculation and misprojection of the commissural axons are indicated by the arrow and arrowhead, respectively. Dashed lines indicate the anterior limb of the anterior commissure. Abbreviations: AC, anterior commissure; ACa, anterior limb of the anterior commissure; ACp, posterior limb of the anterior commissure; CC, corpus callosum; F, fimbria; Fx, fornix; KO, *Shootin1* knockout mouse; MCP, middle cerebellar peduncle; Spt, Septum; VMH, ventromedial hypothalamic nucleus; WT, wild-type mouse. Scale bars: 500 μm.

DOI: https://doi.org/10.7554/eLife.34593.003

The following source data and figure supplements are available for figure 1:

**Figure supplement 1.** Expression and distribution of shootin1a in mouse brain and phenotype of *Shootin1* knockout mice forebrain.
DOI: https://doi.org/10.7554/eLife.34593.004

**Figure supplement 1—source data 1.** Quantitative analyses of the thickness of the corpus callosum, hippocampal commissure and anterior commissure related to *Figure 1—figure supplement 1D*.
DOI: https://doi.org/10.7554/eLife.34593.005

**Figure supplement 2.** Distribution of shootin1a in mouse spinal cord and phenotype of *Shootin1* knockout spinal cord.
DOI: https://doi.org/10.7554/eLife.34593.006

**Figure supplement 2—source data 1.** Quantitative analyses of the thickness of the ventral spinal commissure related to *Figure 1—figure supplement 2C*.
DOI: https://doi.org/10.7554/eLife.34593.007

**Figure supplement 3.** Generation of *Shootin1* knockout mice.
DOI: https://doi.org/10.7554/eLife.34593.008

axon guidance in vitro and in vivo remains unknown. In addition, how shootin1a associates with the substrates through L1-CAM is uncharacterized (*Kubo et al., 2015*).

Here, we combined gene knockout, protein interaction assays, force microscopy, speckle imaging and microfluidics to define a gradient-reading and mechano-effector machinery for netrin-1–induced axon guidance. We show that shootin1a is expressed at high levels in developing forebrain commissural axons and that *Shootin1* knockout mice display abnormal guidance of these axons, a phenotype similar to that of *Netrin-1* knockout mice. Notably, very small spatial differences in netrin-1 concentration elicited highly polarized directional phosphorylation of shootin1a within growth cones. Netrin-1–elicited shootin1a phosphorylation promoted direct interaction between shootin1a and L1-CAM, thereby generating traction force for growth cone motility. Furthermore, disturbance of the spatially regulated shootin1a phosphorylation within growth cones inhibited axon turning, but not axon outgrowth, induced by netrin-1 gradients. Our data demonstrate that shootin1a, through its spatially regulated phosphorylation within growth cones, mediates the gradient reading and mechanoresponse for netrin-1–induced axon guidance.

## Results

### Shootin1a is localized at high levels in axonal tracts of developing mouse brain

To assess a role for shootin1a in axon guidance, we first analyzed its localization in the developing mouse brain. Consistent with a previous report (*Toriyama et al., 2006*), immunoblot analyses detected a low level of shootin1a in embryonic day (E) 13.5 mouse brains (*Figure 1—figure supplement 1A*). The expression increased remarkably between E13.5 and E16.5, remained high through postnatal days (P) 0–12, and then decreased to a low level in the adult. Immunohistochemical analyses with shootin1a-specific antibody detected shootin1a localization widely in P0 brain, with high levels of immunolabeling in the axonal tracts, such as the corpus callosum, anterior commissure,

hippocampal commissure, fornix, fimbria and middle cerebellar peduncle (*Figure 1A* and *Figure 1—figure supplement 1B*). We also detected high levels of shootin1a immunoreactivity that colocalized extensively with that of the axonal marker L1-CAM (*Chung et al., 1991*; *Klingler et al., 2015*) in the corpus callosum and anterior commissure at E16.5 (*Figure 1B*). However, we could not detect shootin1a immunoreactivity in the ventral commissure of the spinal cord at E12 (arrow, *Figure 1—figure supplement 2A*) when high levels of both netrin-1 and DCC are expressed (*Keino-Masu et al., 1996*; *Kennedy et al., 2006*; *Bin et al., 2015*; *Dominici et al., 2017*).

## *Shootin1* knockout mice display abnormal projection of forebrain commissural axons

To analyze further the roles of shootin1a in the developing brain, we generated *Shootin1* knockout mice (*Figure 1—figure supplement 3*). Southern blot analysis confirmed that the first exon of the *Shootin1* gene had been replaced with the β-galactosidase (*LacZ*) and neomycin resistance (Neo$^r$) genes (*Figure 1—figure supplement 3A and B*). Immunoblot analysis demonstrated a complete loss of shootin1a protein (*Figure 1—figure supplement 3C*); shootin1a immunoreactivity was undetectable in *Shootin1* knockout mouse brain sections (*Figure 1—figure supplement 3D*). *Shootin1* knockout mice were born but 13.3% of them died during P0-P20 ($n$ = 98). Coronal and horizontal sections of *Shootin1* knockout brains revealed dysgenesis of the corpus callosum, anterior commissure and hippocampal commissure with a penetrance of 69.2% ($n$ = 26) (*Figure 1C and D* and *Figure 1—figure supplement 1C*); their thicknesses were significantly reduced by *Shootin1* knockout (*Figure 1—figure supplement 1D*). We also observed multiple defects in the brain of *Shootin1* knockout mice, including agenesis of the septum (*Figure 1C*); the detailed analyses of these phenotypes will be reported separately.

To analyze the commissure structures in detail, we visualized axon bundles using anti-L1-CAM antibody. In addition to decreased thickness of the axonal tracts at the midline of the corpus callosum (yellow line, *Figure 1E*), the prominent axonal tracts observed in the intermediate zone of the neocortex of wild-type mice (arrows, *Figure 1E*) were undetectable in *Shootin1* knockout mice. In contrast, ectopic axonal projections toward the cortical plate were observed in *Shootin1* knockout mice (arrowheads, *Figure 1E*). Consistent with the L1-CAM immunolabeling, DiI tracing also showed incomplete contralateral projections of callosal axons (arrowhead, *Figure 1F*). In the anterior commissure, L1-CAM immunolabeling showed a decrease in the thickness of the axon bundle that crosses the midline (yellow line, *Figure 1G*) as well as disruption of the anterior and posterior limbs of the commissure (arrowheads, *Figure 1G*). DiI tracing of the anterior limbs of the anterior commissure demonstrated defasciculation (arrow, *Figure 1H*) and misprojection (arrowhead, *Figure 1H*) of the axons. Consistent with the undetectable level of shootin1a expression in the ventral commissure of the spinal cord (arrow, *Figure 1—figure supplement 2A*), we could not observe noticeable defects of the spinal cord commissure axons in *Shootin1* knockout mice (*Figure 1—figure supplement 2B and C*).

## Shallow gradients of netrin-1 elicit highly polarized shootin1a phosphorylation within growth cones

Extracellular gradients of netrin-1 induce growth cone attraction in vitro (*Serafini et al., 1994*; *Hong et al., 1999*; *Bhattacharjee et al., 2010*). To analyze the growth cone response under netrin-1 gradients, we prepared a device with microjet arrays that can generate gradients of diffusible molecules in the culture medium (*Bhattacharjee et al., 2010*) (*Figure 2A*). The substrates for neuronal culture, glass coverslips, were coated sequentially with polylysine and L1-CAM-Fc as reported (*Shimada et al., 2008*; *Toriyama et al., 2013*; *Kubo et al., 2015*). To estimate the soluble netrin-1 gradients, we used bovine serum albumin (BSA) labeled with the fluorescent tracer Alexa Fluor 594 or Alexa Fluor 488 as a proxy for netrin-1. As the molecular weight of BSA (66.0 kDa) is similar to that of mouse netrin-1 (68.2 kDa), we expect that the gradient of BSA mimics that of netrin-1 in the device. As reported (*Bhattacharjee et al., 2010*), our device generated stable gradients of BSA in the medium (*Figure 2B* and *Video 1*). The difference in the BSA concentrations at the source side end and the other end of the area that expands 400 μm at the center of the linear gradient (red rectangle, *Figure 2A*), estimated by the fluorescence intensity, was 15% (*Figure 2B*). As the regular growth cone width of cultured hippocampal neurons is about 10 μm (*Katsuno et al., 2015*), we

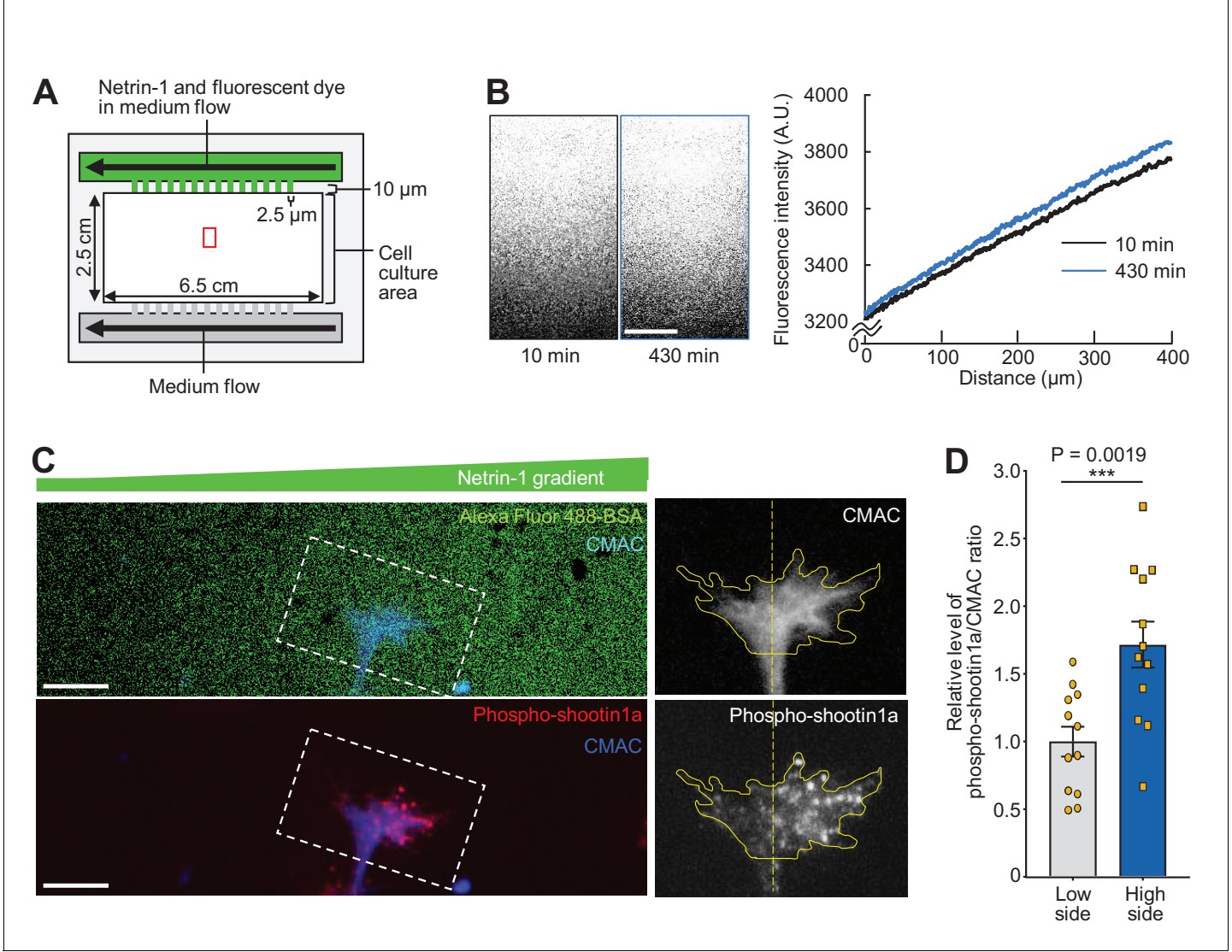

**Figure 2.** Netrin-1 gradients induce asymmetrically localized phosphorylation of shootin1a within single growth cones. (**A**) A schematic diagram of the device with microjet arrays that generates gradients of diffusible molecules in the culture medium. (**B**) Time-lapse fluorescence images of fluorescent dye (Alexa Fluor 488-BSA) in the cell culture area of the device in (A, red rectangle). See *Video 1*. The graph (right) depicts line scans of the fluorescence intensity across the field at 10 min (black line) and 430 min (blue line) during time-lapse imaging. A stable gradient of Alexa Fluor 488-BSA was generated in the device. Bar: 100 µm. (**C**) Neurons cultured in the device were labeled with CMAC (blue) and exposed to gradients of netrin-1 and Alexa Fluor 488-BSA (green) for 30 min. They were then fixed and immunolabeled with an antibody that recognizes shootin1a phosphorylation at Ser249 (red). The right panels show the fluorescent signals of CMAC and phospho-shootin1a in the growth cone located in the corresponding dashed rectangle. Yellow lines and dotted lines indicate the boundary and center line of the growth cone, respectively. A higher level of phospho-shootin1a immunolabeling was observed on the netrin-1 source side. Bar: 10 µm. (**D**) Quantification of relative phospho-shootin1a immunolabeling levels (phosopho-shootin1a immunoreactivity/CMAC staining) in the netrin-1 source side (high side) and control side (low side) of single growth cones. *n* = 12 growth cones. Data represent means ± SEM; ***p<0.01 (unpaired Student's *t*-test).

DOI: https://doi.org/10.7554/eLife.34593.009

The following source data and figure supplements are available for figure 2:

**Source data 1.** Quantification of relative phospho-shootin1a immunolabeling levels related to *Figure 2D*.
DOI: https://doi.org/10.7554/eLife.34593.014
**Figure supplement 1.** Netrin-1 gradients produced on the substrate.
DOI: https://doi.org/10.7554/eLife.34593.010
**Figure supplement 1—source data 1.** Quantitative analyses of the netrin-1 attached to the glass coverslips related to *Figure 2—figure supplement 1B*.
DOI: https://doi.org/10.7554/eLife.34593.011

*Figure 2 continued on next page*

*Figure 2 continued*

**Figure supplement 2.** BSA gradients do not elicit polarized phosphorylation of shootin1 within growth cones.
DOI: https://doi.org/10.7554/eLife.34593.012
**Figure supplement 2—source data 1.** Quantification of relative phospho-shootin1a immunolabeling levels related to *Figure 2—figure supplement 2B*.
DOI: https://doi.org/10.7554/eLife.34593.013

estimate that gradient steepness (the percentage change in concentration) (*Rosoff et al., 2004*) of netrin-1 that covers growth cones located in the red rectangle area (*Figure 2A*) is about 0.4%.

A previous study reported that netrin-1 attaches to polylysine–coated substrates, thereby mediating chemotropic axon guidance (*Moore et al., 2012*). To examine whether the present assay system produces netrin-1 gradients on the substrate, we next analyzed the attachment of netrin-1 to substrates coated with L1-CAM. As reported (*Moore et al., 2012*), incubation of glass coverslips with netrin-1 (200 or 300 ng/ml) led to netrin-1 attachment on the polylysine–coated substrate within 15 min, and the attachment further increased after incubation for 7 hr (*Figure 2—figure supplement 1A and B*). Netrin-1 also attached to the L1-CAM–coated substrate in a manner dependent on the incubation time (*Figure 2—figure supplement 1A and B*). We also confirmed that our device produces a netrin-1 gradient attached to the substrate in a manner dependent on the incubation time and that the difference in concentration across the growth cone is about 0.6% at 10 min and 0.8% at 430 min (*Figure 2—figure supplement 1C*), which is similar to that of BSA (*Figure 2B*). However, the amount of attached netrin-1 was 39% of that on polylysine after 420 min incubation (*Figure 2—figure supplement 1A and B*), suggesting that at least 61% of the applied netrin-1 is not attached to the substrate under our conditions. Together, these results indicate that gradients of both soluble and substrate-bound netrin-1 are produced in our culture system.

We previously reported that netrin-1 induces Pak1–mediated phosphorylation of shootin1a at Ser101 and Ser249 in axonal growth cones (*Toriyama et al., 2013*). Using the microjet device, we examined the localization of netrin1–elicited shootin1a phosphorylation in growth cones. Hippocampal neurons cultured in the device for 1.5–2 days were labeled with the fluorescent volume marker 7-amino-4-chloromethylcoumarin (CMAC) and exposed to a netrin-1 gradient for 30 min. The neurons were then fixed and immunolabeled with an antibody that recognizes shootin1a phosphorylation at Ser249. We selected axons located near the center of the gradients (red rectangle, *Figure 2A*) and that were oriented approximately perpendicular to the netrin-1 gradient. To our surprise, quantification of the phospho-shootin1a immunoreactivity and CMAC staining revealed a highly polarized localization of the phosphorylated shootin1a within growth cones (*Figure 2C*). The relative level of the phosphorylated shootin1a (phosopho-shootin1a immunoreactivity/CMAC staining) was 71% higher on the netrin-1 source side than on the control side (p=0.002, *n* = 12) (*Figure 2D*), and contrasted markedly with the shallow gradients of extracellular netrin-1 estimated by the fluorescent tracer (*Figure 2B*) and antibody (*Figure 2—figure supplement 1C*). On the other hand, similar gradients of the control molecule BSA in the medium did not elicit polarized phosphorylation of shootin1 within growth cones (*Figure 2—figure supplement 2*).

## Netrin-1–induced shootin1a phosphorylation promotes shootin1a–L1-CAM interaction

We reported previously that netrin-1–induced shootin1a phosphorylation at Ser101 and Ser249 enhances the coupling between F-actin retrograde flow and L1-CAM at growth cones (*Toriyama et al., 2013*). However, whether shootin1 interacts directly with L1-CAM as well as whether this interaction is regulated by the netrin-1 signaling are unclear (*Kubo et al., 2015*). To clarify these points, we performed an in vitro binding assay using purified shootin1a and GST–tagged intracellular domain (ICD) (1145-1257 a.a.) of L1-CAM. As shown in *Figure 3A*, shootin1a directly interacted with L1-CAM-ICD.

To determine whether the shootin1a phosphorylation modulates the shootin1a–L1-CAM interaction, we analyzed the interaction between the phosphorylated shootin1a and L1-CAM-ICD. An in vitro binding assay with purified proteins showed that the interaction of L1-CAM-ICD with phospho-mimic shootin1a (shootin1a-DD), in which Ser101 and Ser249 were replaced by aspartate, was stronger than that with wild-type shootin1a (shootin1a-WT) (*Figure 3B*): the apparent dissociation

constant for shootin1a-DD ($K_d$ = 46.4 ± 4.4 nM) was 2.9-fold lower (p<0.02, $n$ = 3 independent experiments) than that of shootin1a-WT ($K_d$ = 133.5 ± 13.6 nM) (*Figure 3C*). In vitro phosphorylation and binding assays also demonstrated that phosphorylation of shootin1a by Pak1 promotes its interaction with L1-CAM-ICD (*Figure 3D and E*). Ectopic expression of a constitutively active Pak1 in HEK293T cells increased phosphorylation of myc-shootin1a at Ser101 and Ser249 in these cells (*Figure 4A and B*); this in turn promoted the interaction between shootin1a and L1-CAM-ICD (*Figure 4A and B*). Conversely, expression of a dominant-negative Pak1 decreased shootin1a phosphorylation and inhibited the interaction (*Figure 4A and B*). Furthermore, stimulation of neurons by netrin-1 increased the phosphorylation of shootin1a at Ser101 and Ser249 (*Figure 4C and D*); this led to a concomitant increase in the interaction between endogenous shootin1a and L1-CAM (*Figure 4C and D*). In axonal growth cones, phosphorylated shootin1a was highly colocalized with L1-CAM (*Figure 4E*). Altogether, our data demonstrate that netrin-1–induced shootin1a phosphorylation promotes direct interaction between shootin1a and L1-CAM.

## Shootin1a–L1-CAM interaction mediates netrin-1–induced F-actin–adhesion coupling and mechanoresponse

To address the role of the shootin1a–L1-CAM interaction, we next analyzed the shootin1a region that interacts with L1-CAM. An in vitro binding assay with purified proteins showed that residues 1–125 of shootin1a (shootin1a (1-125)) were essential and sufficient to bind to L1-CAM (*Figure 5A*). On the other hand, we previously reported that another region, shootin1a (261-377), is responsible for its interaction with cortactin, which links shootin1a to F-actin flow (*Figure 5A*) (*Kubo et al., 2015*). Consistent with these data, when expressed in hippocampal neurons, myc-shootin1a (1-125) was highly colocalized with L1-CAM in axonal growth cones (*Figure 5B*). As shootin1a (1-125) interacts with L1-CAM but not with cortactin, we expected that it can be used as a dominant negative mutant that disrupts the shootin1a–L1-CAM interaction. Indeed, shootin1a (1-125) overexpressed in HEK293T cells bound to L1-CAM-ICD, and inhibited the interaction between shootin1a and L1-CAM-ICD (*Figure 5C*).

Coupling between F-actins and substrate reduces the speed of F-actin flow in growth

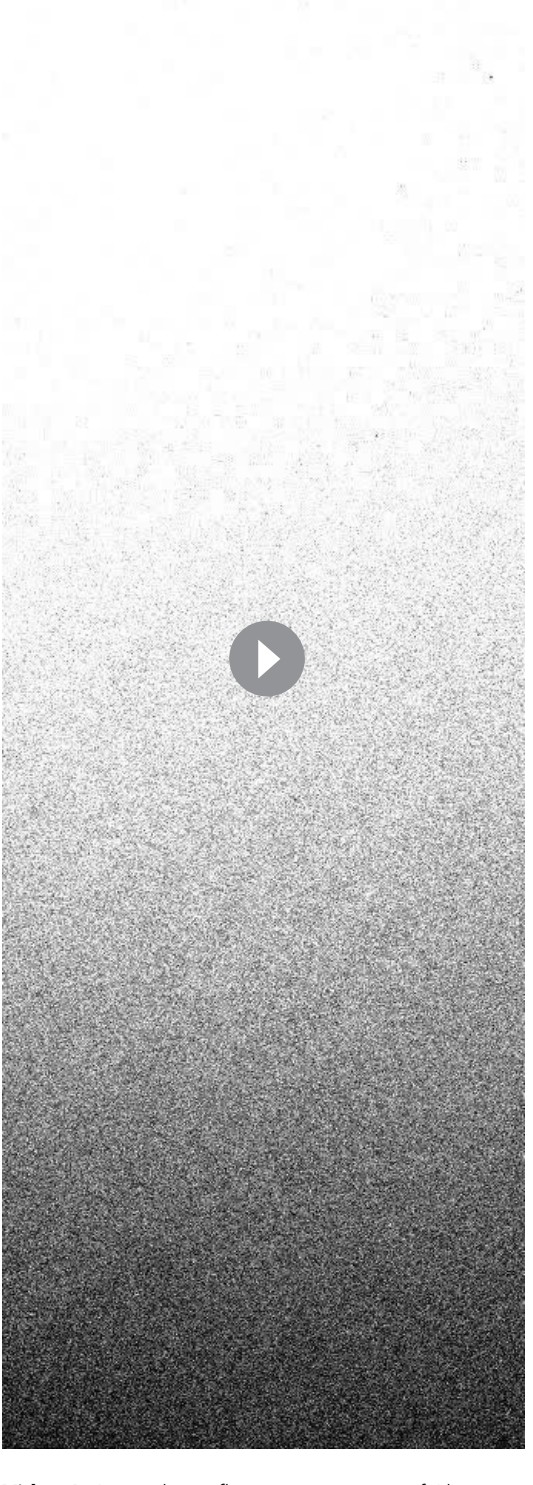

**Video 1.** A time-lapse fluorescence movie of Alexa Fluor 488-BSA in the cell culture area of the device in (*Figure 2A*, red rectangle). See the legend for *Figure 2B*.
DOI: https://doi.org/10.7554/eLife.34593.015

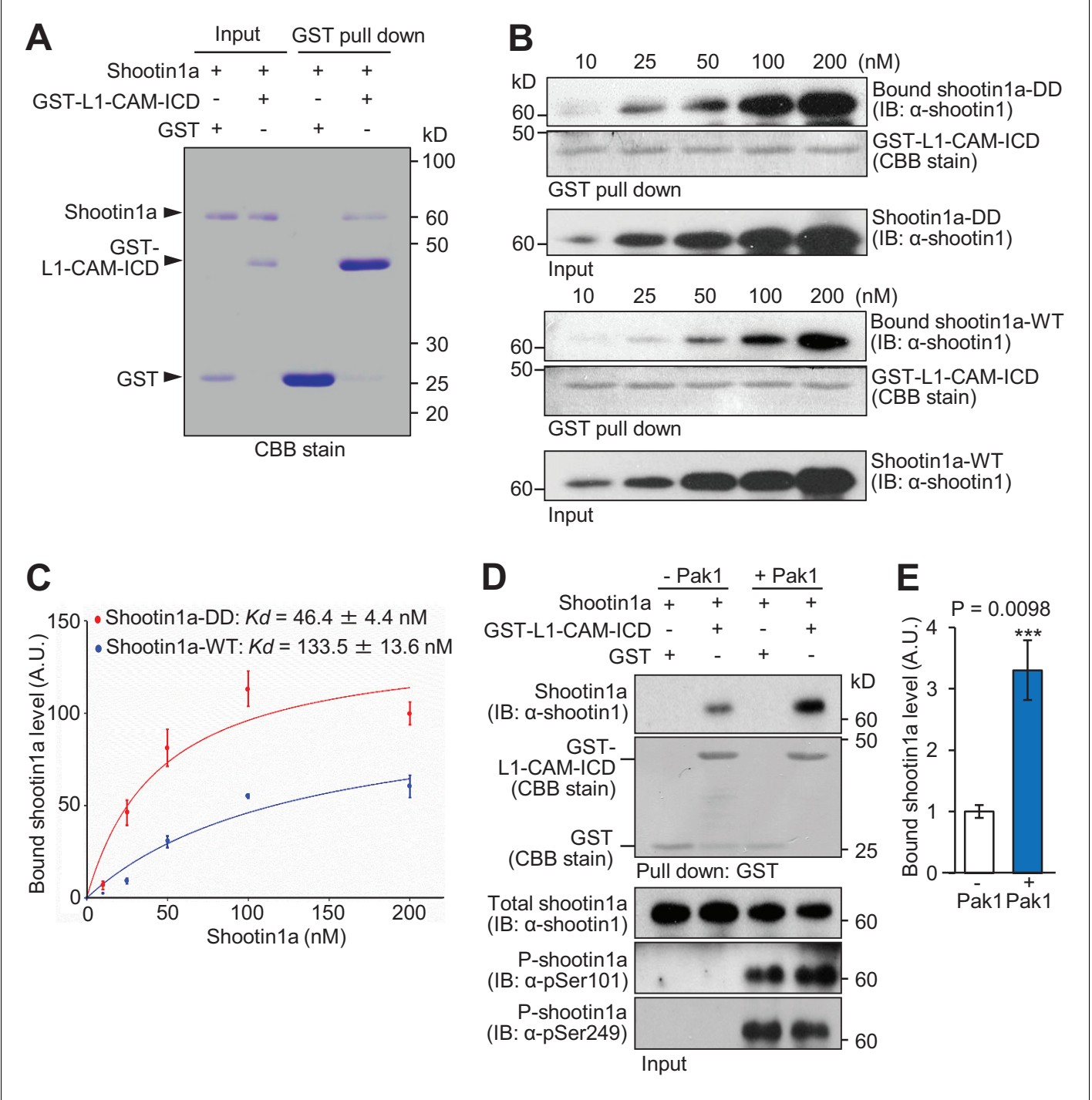

**Figure 3.** Pak1-mediated shootin1a phosphorylation enhances the interaction between shootin1a and L1-CAM. (**A**) In vitro binding assay using purified shootin1a-WT (100 nM) and purified GST-L1-CAM-ICD (100 nM). Proteins were incubated with Glutathione Sepharose 4B and GST-L1-CAM-ICD was eluted. The eluate was then analyzed by SDS-PAGE and CBB staining; 0.2% of the input proteins were also analyzed. (**B** and **C**) In vitro binding assay using purified shootin1a-WT or purified shootin1a-DD and purified GST-L1-CAM-ICD. Shootin1a-DD or shootin1a-WT at increasing concentrations was incubated with GST-L1-CAM-ICD and Glutathione Sepharose 4B. GST-L1-CAM-ICD was eluted. After SDS-PAGE, the eluate was immunoblotted with anti-shootin1 antibody or stained with CBB (**B**), and the bound shootin1a-DD and shootin1a-WT were then quantified (**C**). Data represent means ± SEM (*n* = 3 independent experiments). (**D** and **E**) In vitro binding assay using Pak1-phosphorylated purified shootin1a and purified GST-L1-CAM-ICD. Shootin1a-WT (100 nM) or Pak1-phosphorylated shootin1a-WT (100 nM) was incubated with GST-L1-CAM-ICD and Glutathione Sepharose 4B. GST-L1-CAM-ICD was eluted. After SDS-PAGE, the eluate was immunoblotted with anti-shootin1 antibody or stained with CBB (**D**). Input proteins (1%) were

*Figure 3 continued on next page*

*Figure 3 continued*

also analyzed with anti-shootin1, anti-pSer101-shootin1 or anti-pSer249-shootin1 antibody. Quantitative data for bound shootin1a are shown in (E) (*n* = 3 independent experiments). Data represent means ± SEM; ***p<0.01 (unpaired Student's *t*-test).

DOI: https://doi.org/10.7554/eLife.34593.016

The following source data is available for figure 3:

**Source data 1.** Quantitative data for Kd value related to *Figure 3C*.
DOI: https://doi.org/10.7554/eLife.34593.017

**Source data 2.** Quantitative data for bound shootin1a related to *Figure 3E*.
DOI: https://doi.org/10.7554/eLife.34593.018

cones (*Suter et al., 1998*; *Toriyama et al., 2013*). Using shootin1a (1-125) as a dominant negative mutant, we examined whether the shootin1a–L1-CAM interaction is involved in netrin-1–induced mechanical coupling between F-actin flow and the substrate. Hippocampal neurons expressing mRFP-actin were cultured on coverslips coated with L1-CAM, and F-actin flow in the growth cone was monitored by live-cell fluorescence microscopy (*Figure 6A*, *Video 2*). In neurons overexpressing a control protein myc-GST, the fluorescent features of mRFP-actin moved retrogradely at 4.5 ± 0.1 µm/min (mean ± SE, *n* = 30 fluorescent features), as previously reported (*Shimada et al., 2008*). Overexpression of myc-shootin1a (1-125) in hippocampal neurons increased the velocity of F-actin flow (*Figure 6A and B*), suggesting that shootin1a (1-125) inhibits the F-actin–adhesion coupling. Netrin-1 stimulation significantly decreased the velocity of F-actin flow in control growth cones, reflecting promotion of the F-actin–adhesion coupling. On the other hand, netrin-1 accelerated the flow in the presence of overexpressed myc-shootin1a (1-125) (*Figure 6A and B*); this can be explained by the inhibition of the F-actin adhesion coupling as well as a simultaneous increase in actin polymerization by netrin-1–induced activation of Cdc42 and Rac1 (*Shekarabi et al., 2005*; *Briançon-Marjollet et al., 2008*). These results suggest that shootin1a–L1-CAM interaction mediates netrin-1–induced F-actin–adhesion coupling.

We further monitored mechanoresponses of growth cones using traction force microscopy. Hippocampal neurons were cultured on L1-CAM–coated polyacrylamide gels with embedded 200 nm fluorescent beads. Traction forces under the growth cones were monitored by visualizing force–induced deformation of the elastic substrate, which is reflected by displacement of the beads from their original positions. As reported (*Toriyama et al., 2013*), the reporter beads under the growth cones moved dynamically, reflecting the traction force on the substrate (*Figure 6C*, *Video 3*); and the force was increased significantly by netrin-1 stimulation (*Figure 6C and D* and *Figure 6—figure supplement 1A*). Importantly, inhibition of the shootin1a–L1-CAM interaction by overexpressing myc-shootin1a (1-125) significantly decreased traction forces and abolished the netrin-1–induced increase in traction forces (*Figure 6D*). Overexpression of myc-shootin1a (1-125) also decreased axon length and abolished the netrin-1–induced axon outgrowth (*Figure 6—figure supplement 1B* and *Figure 6E*), suggesting that shootin1a–L1-CAM interaction is involved in the netrin-1–induced generation of traction force for growth cone migration.

## Netrin-1–induced axon attraction requires shootin1a

Next, we examined whether shootin1a is involved in netrin-1–induced axon guidance. Hippocampal neurons were stimulated with a netrin-1 gradient for 7 hr; the gradient was applied approximately perpendicularly to the extending direction of axons (*Figure 7—figure supplement 1A*). The right panel (*Figure 7—figure supplement 1A*) depicts the migration of individual axonal growth cones. Consistent with previous reports (*Kennedy et al., 1994*; *Serafini et al., 1994*; *Bhattacharjee et al., 2010*), the majority of the axonal growth cones migrated toward the netrin-1 source (*Figure 7—figure supplement 1A*, *Video 4*). The mean axon outgrowth velocity was 24.4 ± 0.7 µm/h (*Figure 7—figure supplement 1B*), and the net change in the angle of the growth cone toward the netrin-1 source was 26.4 ± 0.1° (*n* = 9) (*Figure 7—figure supplement 1C*). As our assay system produces gradients of both soluble and substrate-bound netrin-1 (*Figure 2B* and *Figure 2—figure supplement 1C*), we assessed the abilities of soluble and substrate-bound netrin-1 to turn axons, by solubilizing netrin-1 with heparin. As in the case of polylysine–coated substrate (*Moore et al., 2012*), inclusion of 2 µg/ml heparin in the culture medium released netrin-1 from the L1-CAM–coated substrate

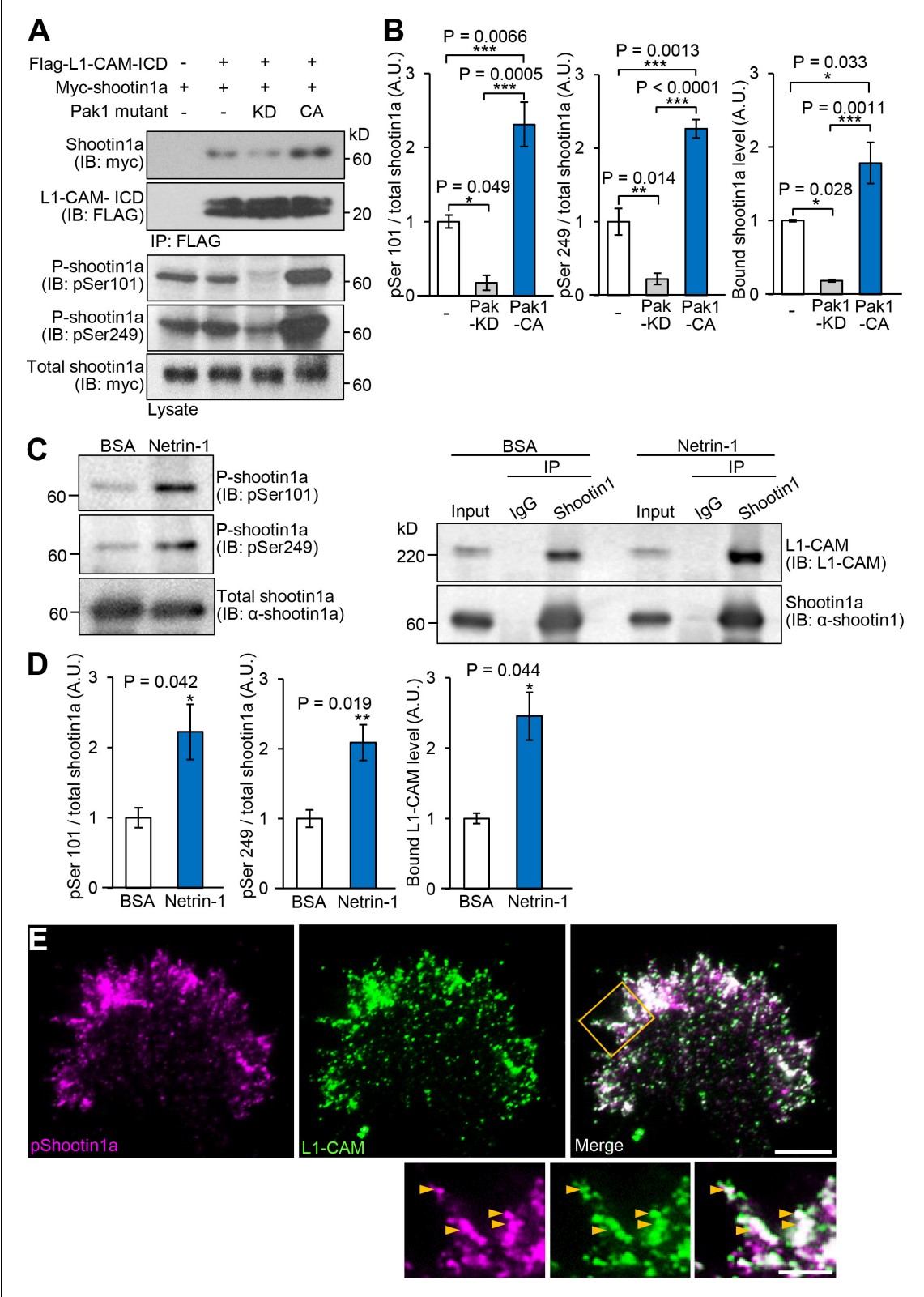

**Figure 4.** Netrin-1–induced Pak1-mediated shootin1a phosphorylation enhances the interaction between shootin1a and L1-CAM. (**A** and **B**) Co-immunoprecipitation of myc-shootin1a and FLAG-L1-CAM-ICD in HEK293T cells. Cells were transfected with vectors to express myc-shootin1a and FLAG-L1-CAM-ICD; some of them were also co-transfected with a vector to express dominant negative Pak1 (KD) or constitutively active Pak1 (CA) as indicated. Cell lysates were then incubated with anti-FLAG antibody. The immunoprecipitates were immunoblotted with anti-myc or anti-FLAG

*Figure 4 continued on next page*

*Figure 4 continued*

antibody (**A**). Cell lysates (1%) were also analyzed with anti-pSer101-shootin1, anti-pSer249-shootin1, or anti-myc antibody. Quantitative data for phosphorylated and bound shootin1a are shown in (**B**) (*n* = 3 independent experiments). Data represent means ± SEM; ***p<0.01; **p<0.02; *p<0.05 (One-way ANOVA with Tukey's post hoc test). (**C** and **D**) Co-immunoprecipitation of shootin1a and L1-CAM in cultured cortical neurons. After incubation of neurons with 4.4 nM netrin-1 or BSA (control) for 1 hr, cell lysates were prepared and incubated with anti-shootin1 antibody (right panel). The immunoprecipitates were immunoblotted with anti-shootin1 or anti-L1-CAM antibody. The cell lysates (5%) were also analyzed with anti-pSer101-shootin1, anti-pSer249-shootin1, or anti-shootin1a antibody (left panel). Quantitative data for phosphorylated shootin1a and bound L1-CAM are shown in (**D**) (*n* = 3 independent experiments). Data represent means ± SEM; **p<0.02; *p<0.05 (Unpaired Student's *t*-test). (**E**) Fluorescence images of an axonal growth cone labeled with anti-pSer249-shootin1a (magenta) and anti-L1-CAM (green) antibodies. The cells were observed using a TIRF microscope. An enlarged view of the filopodium in the rectangle is shown in the lower panel. Arrowheads indicate phosphorylated shootin1a colocalized with L1-CAM. Bar: 5 µm (in the inset, 2 µm).

DOI: https://doi.org/10.7554/eLife.34593.019

The following source data is available for figure 4:

**Source data 1.** Quantitative data for phosphorylated and bound shootin1a related to *Figure 4B*.
DOI: https://doi.org/10.7554/eLife.34593.020
**Source data 2.** Quantitative data for phosphorylated shootin1a and bound L1-CAM related to *Figure 4D*.
DOI: https://doi.org/10.7554/eLife.34593.021

(*Figure 2—figure supplement 1A and B*). In contrast to the data obtained with spinal cord neurons (*Moore et al., 2012*), the netrin-1 gradient induced axon turning even in the presence of heparin, indicating that a gradient of soluble netrin-1 contributes to axon turning of hippocampal neurons (*Figure 7—figure supplement 1D*, *Video 5*). However, the degree of netrin-1–induced axon outgrowth and turning was reduced in the presence of heparin (*Figure 7—figure supplement 1B and C*). These data are consistent with a previous report (*Mai et al., 2009*) that netrin-1 attached to the substrate induces axon turning of cultured hippocampal neurons. Thus, we conclude that gradients of both soluble and substrate-bound netrin-1 contribute to axon turning of hippocampal neurons in our assay system.

We next stimulated hippocampal neurons expressing control miRNA or shootin1a miRNA, which inhibits shootin1a expression, with a netrin-1 gradient. The majority of the axonal growth cones of control neurons migrated toward the netrin-1 source (*Figure 7A*, *Video 6*). The mean axon outgrowth velocity was 26.1 ± 3.0 µm/h (*Figure 7C*), and the net change in the angle of the growth cone toward the netrin-1 source was 32.7 ± 2.2° (*n* = 11) (*Figure 7D*). Repression of shootin1a by RNAi not only reduced the axon outgrowth velocity (5.6 ± 0.9 µm/h, p<0.01) (*Figure 7B and C*, *Video 7*) but also inhibited the growth cone turning toward the netrin-1 source (2.1 ± 1.2°, p<0.01, *n* = 13) (*Figure 7B and D*). Similar results were obtained using hippocampal neurons prepared from *Shootin1* knockout mice (*Figure 7—figure supplement 2*). Together, these data indicate that netrin-1–induced axon guidance of hippocampal neurons on an L1-CAM substrate requires shootin1a.

## Netrin-1–induced axon attraction requires shootin1a–L1-CAM interaction

We further examined a role of shootin1a–L1-CAM interaction in netrin-1–induced axon guidance. Hippocampal neurons overexpressing EGFP (control) or EGFP-shootin1a (1-125), which inhibits the shootin1a–L1-CAM interaction (*Figure 5C*), were stimulated with a netrin-1 gradient for 7 hr (*Figure 7E and F*, left panels). The majority of the axonal growth cones of control neurons migrated toward the netrin-1 source (*Figure 7E*, *Video 8*). The mean axon outgrowth velocity was 32.4 ± 6.6 µm/h (*Figure 7—figure supplement 3A*), and the net change in the angle of the growth cone toward the netrin-1 source was 28.8 ± 3.8° (*n* = 17) (*Figure 7—figure supplement 3B*). On the other hand, inhibition of the shootin1a–L1-CAM interaction by overexpressing EGFP-shootin1a (1-125) not only reduced the axon outgrowth velocity (13.8 ± 3.4 µm/h, p<0.05) but also inhibited the growth cone turning toward the netrin-1 source (0.6 ± 1.1°, p<0.01, *n* = 16) (*Figure 7F*, *Figure 7—figure supplement 3* and *Video 9*). These data suggest that the interaction between shootin1a and L1-CAM mediates netrin-1–induced axon guidance.

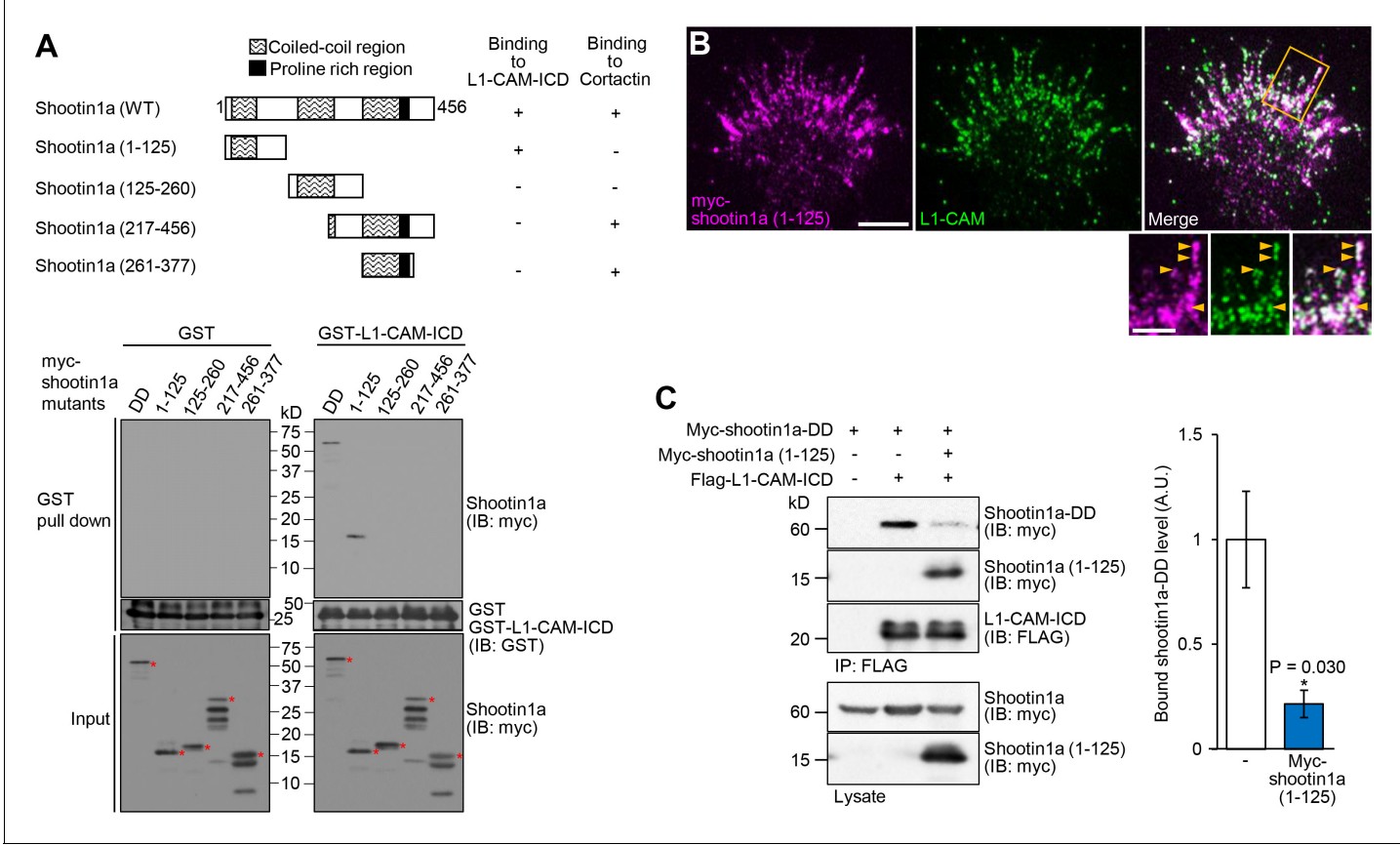

**Figure 5.** Shootin1a (1-125) interacts with L1-CAM and disturbs the interaction between shootin1a and L1-CAM. (**A**) Upper panel: schematic representation of shootin1a (WT) and shootin1a deletion mutants, and their ability to interact with L1-CAM-ICD and cortactin. Lower panel: in vitro binding assay using purified myc-tagged shootin1a mutants and purified GST-L1-CAM-ICD. Myc-shootin1a mutants (100 nM) were incubated with GST-L1-CAM-ICD (100 nM) and Glutathione Sepharose 4B. GST-L1-CAM-ICD was eluted. After SDS-PAGE, the eluate was immunoblotted with anti-myc or anti-GST antibody. Asterisks denote myc shootin1a mutants. (**B**) Neurons transfected with myc-shootin1a (1-125) were labeled with anti-myc (magenta) and anti-L1-CAM (green) antibodies. The cells were observed using a TIRF microscope. An enlarged view of the filopodium in the rectangle is shown in the inset. Arrowheads indicate shootin1a (1-125) colocalized with L1-CAM. Bar: 5 μm (in the inset, 2 μm). (**C**) Overexpressed shootin1a (1-125) inhibits the interaction between shootin1a and L1-CAM-ICD. HEK293T cells were transfected with vectors to express myc-shootin1a and FLAG-L1-CAM-ICD; some of them were also co-transfected with a vector to overexpress myc-shootin1a (1-125) as indicated. Cell lysates were prepared and incubated with anti-FLAG antibody. The immunoprecipitates were immunoblotted with anti-myc or anti-FLAG antibody. The cell lysates (1%) were also analyzed with anti-myc antibody. The graph (right) shows quantitative data for bound shootin1a-DD ($n$ = 3 independent experiments). Data represent means ± SEM; *$p<0.05$ (unpaired Student's $t$-test).

DOI: https://doi.org/10.7554/eLife.34593.022

The following source data is available for figure 5:

**Source data 1.** Quantitative data for bound shootin1a-DD related to *Figure 5C*.

DOI: https://doi.org/10.7554/eLife.34593.023

## Shootin1a–L1-CAM interaction mediates netrin-1–induced axon guidance on laminin

Laminins are widely used substrates for axon guidance assays (*Turney and Bridgman, 2005*; *Nichol et al., 2016*); L1-CAM on growth cones interacts directly with laminin presented on the substrate (*Hall et al., 1997*; *Abe et al., 2018*). To examine whether the shootin1a–L1-CAM interaction mediates netrin-1–induced axon guidance generally, we performed an axon guidance assay on an alternative substrate, laminin. Growth cones of cultured hippocampal neurons on laminin turned in response to netrin-1 gradients (*Figure 7—figure supplement 4A–C*), as they did on L1-CAM (*Figure 7—figure supplement 1A*). To assess shootin1a-mediated clutch coupling on laminin, we measured F-actin retrograde flow in growth cones on laminin. Consistent with previously reported

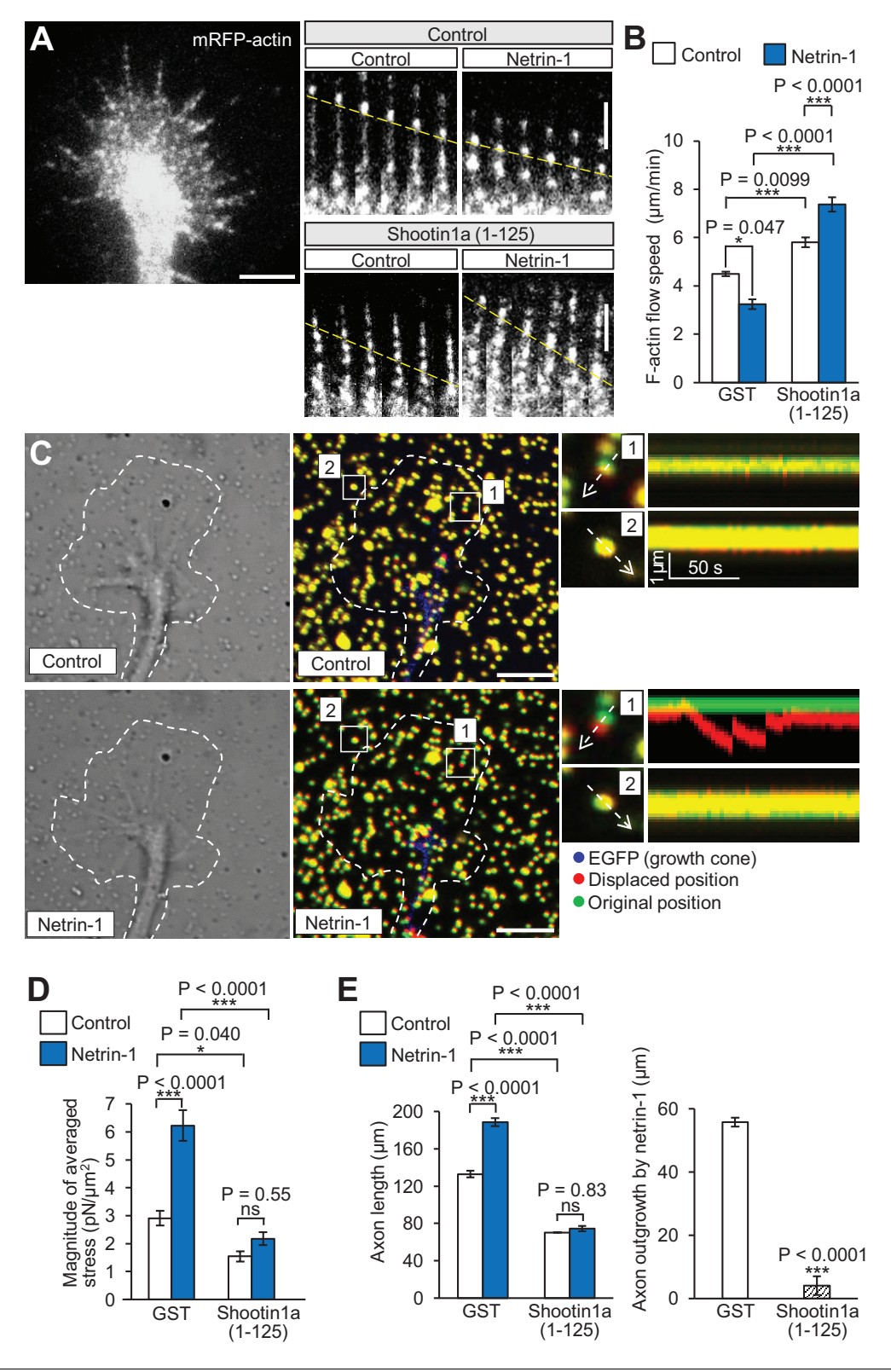

**Figure 6.** Shootin1a–L1-CAM interaction mediates netrin-1–induced F-actin adhesion coupling and mechanoresponse for axon outgrowth. (A) Fluorescent feature images of mRFP-actin at axonal growth cones overexpressing myc-GST (control) or myc-shootin1a (1-125) in the absence (control) or presence of 4.4 nM netrin-1 (see *Video 2*). Kymographs of the fluorescent features of mRFP-actin in filopodia at 5 s intervals are shown (F-actin flows are indicated by dashed yellow lines). (B) F-actin retrograde flow speed measured from the kymograph analysis in A; 120 fluorescent features (47 growth

*Figure 6 continued on next page*

*Figure 6 continued*

cones) were analyzed. One-way ANOVA with Tukey's post hoc test was used. (C) DIC and fluorescence images (left panel) showing an axonal growth cone of a DIV2 neuron overexpressing EGFP and cultured on L1-CAM–coated polyacrylamide gel with embedded 200 nm fluorescent beads. The panels show representative images from time-lapse series taken every 3 s for 150 s before (control) and 60 min after netrin-1 (4.4 nM) stimulation (see *Video 3*). The original and displaced positions of the beads in the gel are indicated by green and red colors, respectively. Dashed lines indicate the boundary of the growth cone. The kymographs (right panel) along the axis of bead displacement (white dashed arrows) at the indicated areas 1 and 2 of the growth cone show movement of beads recorded every 3 s. The bead in area two is a reference bead. (D) Analyses of the magnitude of the traction forces under axonal growth cones overexpressing myc-GST (control) or myc-shootin1a (1-125) before (control) or after netrin-1 stimulation (see *Figure 6—figure supplement 1A* for the direction of the traction forces, n = 14 growth cones). One-way ANOVA with Tukey's post hoc test was performed. (E) Three hours after plating, hippocampal neurons overexpressing myc-GST (control) or myc-shootin1a (1-125) were incubated with BSA (control) or 4.4 nM netrin-1 for 40 hr, and then immunolabeled by anti-myc antibody (see *Figure 6—figure supplement 1B*). Axon length was then analyzed (n = 909 neurons). One-way ANOVA with Schaffer's post hoc test was performed in the left graph, while an unpaired Student's *t*-test was used in the right graph. Data represent means ± SEM; ***p<0.01; *p<0.05; ns, not significant. Bars: 5 μm (in the kymographs of A, 2 μm).
DOI: https://doi.org/10.7554/eLife.34593.024

The following source data and figure supplements are available for figure 6:

**Source data 1.** Quantitative data for F-actin retrograde flow speed related to *Figure 6B*.
DOI: https://doi.org/10.7554/eLife.34593.027
**Source data 2.** Quantitative data for the magnitude of the traction forces related to *Figure 6D*.
DOI: https://doi.org/10.7554/eLife.34593.028
**Source data 3.** Quantitative data for axon length and axon outgrowth by netrin-1 related to *Figure 6E*.
DOI: https://doi.org/10.7554/eLife.34593.029
**Figure supplement 1.** Shootin1a–L1-CAM interaction mediates netrin-1–induced axon outgrowth.
DOI: https://doi.org/10.7554/eLife.34593.025
**Figure supplement 1—source data 1.** Statistical analyses of the angle (°) of the traction forces related to *Figure 6—figure supplement 1A*.
DOI: https://doi.org/10.7554/eLife.34593.026

data (*Abe et al., 2018*), the F-actin retrograde flow rate in control growth cones on laminin was 2.3 ± 0.3 μm/min (*Figure 7—figure supplement 4D*). As in the case of growth cones on L1-CAM (*Figure 6A and B*), overexpression of shootin1a (1-125) increased significantly the retrograde flow rate under these conditions (*Figure 7—figure supplement 4D*), indicating that inhibition of the shootin1a–L1-CAM interaction also disrupts F-actin-adhesion coupling in growth cones on laminin. Furthermore, uncoupling of F-actin-adhesion coupling by shootin1a (1-125) inhibited netrin-1–induced axon outgrowth and turning on laminin (*Figure 7—figure supplement 4A–C and E*). Together, these data indicate that netrin-1–induced axon guidance, which is mediated by the shootin1a–L1-CAM interaction, is not limited to growth cones on L1-CAM.

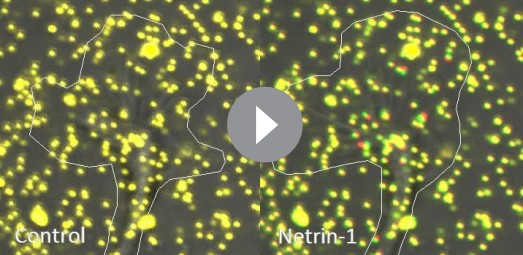

**Video 2.** Movement of fluorescent features of mRFP-actin in a growth cone of a neuron overexpressing myc-GST. See the legend for *Figure 6A*.
DOI: https://doi.org/10.7554/eLife.34593.030

**Video 3.** Netrin-1-induced promotion of traction forces at an axonal growth cone. Left and right panels show bead displacement before and 60 min after netrin-1 stimulation, respectively. See the legend for *Figure 6C*.
DOI: https://doi.org/10.7554/eLife.34593.031

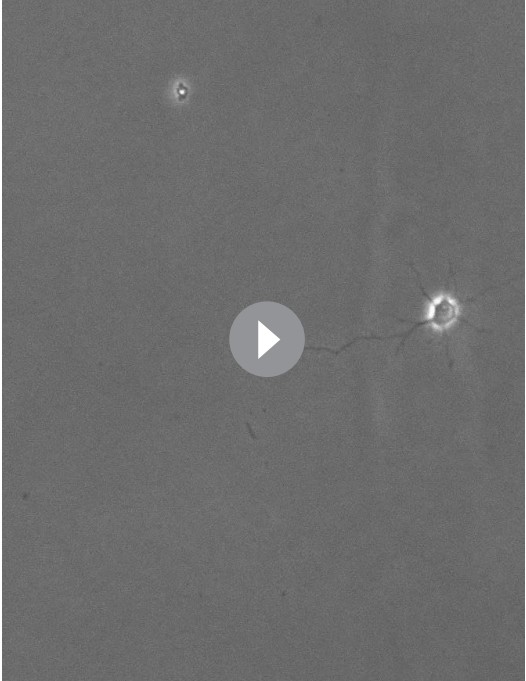

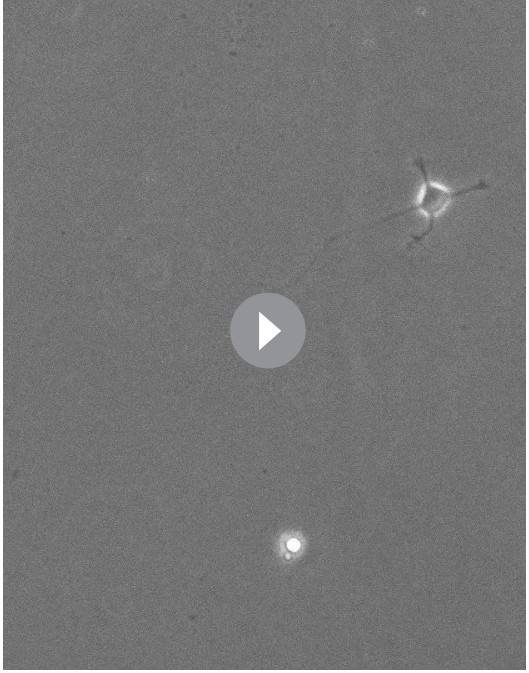

**Video 4.** A time-lapse movie of a hippocampal neuron under a gradient of netrin-1 without heparin. See the legend for *Figure 7—figure supplement 1A*. The gradient of Alexa Fluor 594-BSA in *Figure 7—figure supplement 1A* is not shown.
DOI: https://doi.org/10.7554/eLife.34593.032

**Video 5.** A time-lapse movie of a hippocampal neuron under a gradient of netrin-1 with 2 μg/ml heparin. See the legend for *Figure 7—figure supplement 1D*. The gradient of Alexa Fluor 594-BSA in *Figure 7—figure supplement 1D* is not shown.
DOI: https://doi.org/10.7554/eLife.34593.033

## Netrin-1–induced axon attraction requires polarized shootin1a phosphorylation within growth cones

Finally, we analyzed the role of polarized shootin1 phosphorylation within growth cones elicited by netrin-1 gradients. As shown above (*Figure 7A–D*), repression of shootin1a by expression of shootin1a miRNA inhibited axon outgrowth and growth cone turning toward the netrin-1 source. Expression of RNAi-refractory shootin1a-WT in neurons expressing shootin1a miRNA rescued the reduction of axon outgrowth as well as growth cone turning (*Figure 8A, C and D*, *Video 10*), indicating that shootin1a regulates both axon outgrowth and growth cone turning. Our previous work has shown that axon outgrowth is regulated by shootin1a–mediated clutch coupling (*Shimada et al., 2008*; *Kubo et al., 2015*). As shootin1a-DD, the constitutively active shootin1a, mediates clutch coupling and force generation in the absence of PAK1 activity (*Toriyama et al., 2013*) but cannot be regulated by phosphorylation, displacement of wild-type shootin1a with shootin1a-DD would disturb netrin-1–induced polarized shootin1a regulation without disturbing the clutch coupling. As shown by *Figure 8B–D* and *Video 11*, disturbance of polarized shootin1a phosphorylation within growth cones by this displacement inhibited growth cone turning toward the netrin-1 source without reducing axon outgrowth velocity. These results demonstrate that the disturbance of axon turning caused by depletion of shootin1a (*Figure 7D* and *Figure 7—figure supplement 2D*) or by dominant negative shootin1a (*Figure 7—figure supplement 3B*) is not attributed only to the inhibited axon outgrowth, and suggest that the polarized phosphorylation of shootin1a within growth cones is required for the directional axon guidance induced by netrin-1 gradients (*Figure 8E*).

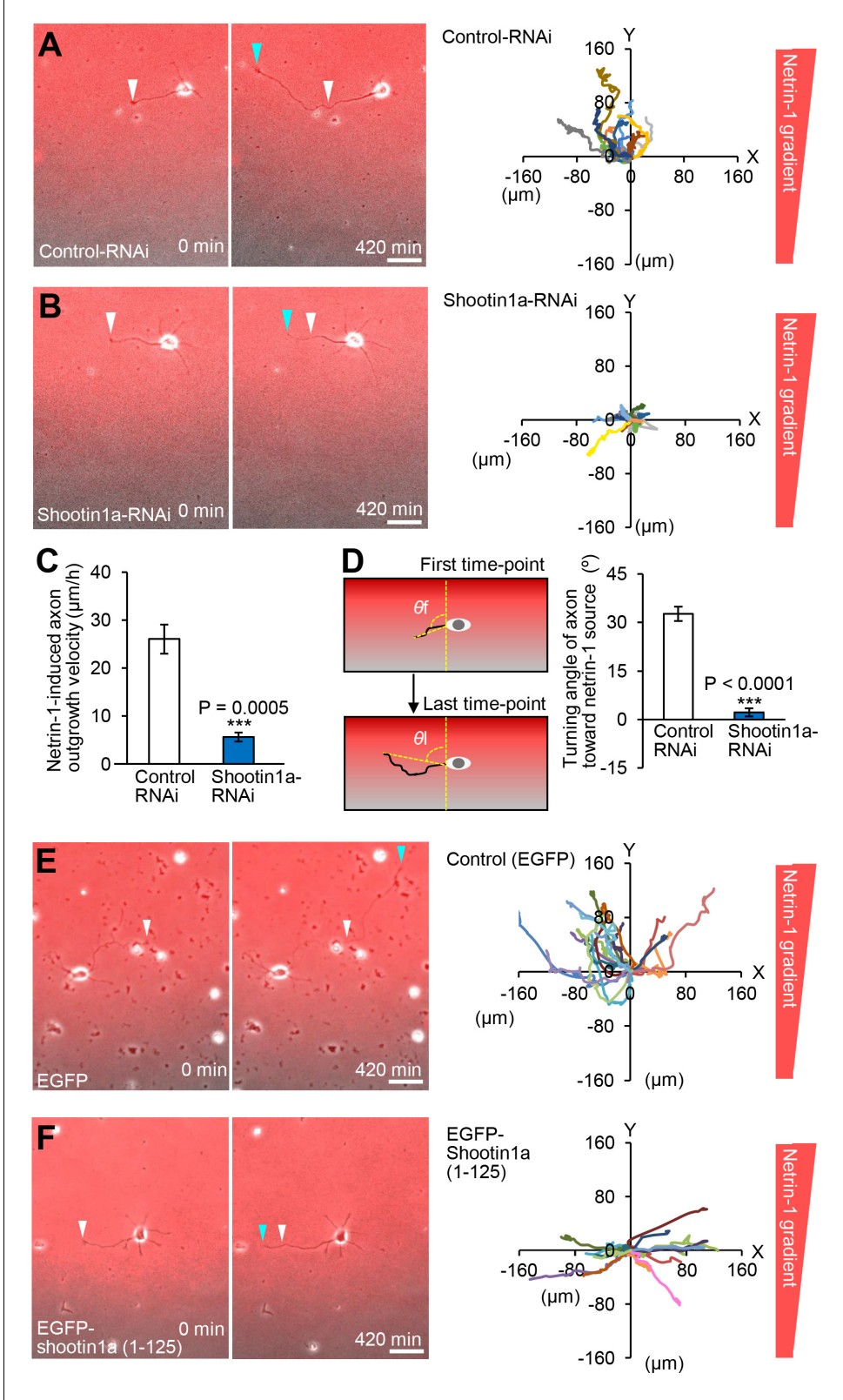

**Figure 7.** Shootin1a and shootin1a–L1-CAM interaction mediate netrin-1–induced axon guidance. (A and B) Time-lapse phase-contrast/fluorescence images of hippocampal neurons expressing control miRNA (A) and shootin1a miRNA (B) under the gradients of netrin-1 and Alexa Fluor 594-BSA. White and blue arrowheads indicate growth cones at the first and last time-points, respectively. See **Videos 6** and **7**. The right panels depict trajectories of individual growth cone migrations. The initial growth cone positions are normalized at (x = 0 μm, y = 0 μm). Bars: 50 μm. (C) Axon

*Figure 7 continued on next page*

*Figure 7 continued*

outgrowth velocity obtained from the analyses in (**A** and **B**) (*n* = 24 growth cones). See also the legend for *Figure 8C* about quantitative data. (**D**) Turning angle of axon toward the netrin-1 source was obtained from the analyses in (**A** and **B**), by calculating the difference between the angles of the axonal tip at the first and last time-points of the observations (*θ*f - *θ*l). The graph shows quantified data (*n* = 24 growth cones). See also the legend for *Figure 8D* about quantitative data. (**E** and **F**) Time-lapse phase-contrast/fluorescence images of hippocampal neurons expressing EGFP (control) (**E**) and EGFP-shootin1a (1-125) (**F**) under gradients of netrin-1 and Alexa Fluor 594-BSA (red). White and blue arrowheads indicate growth cones at the first and last time-points, respectively. See *Videos 8* and *9*. The right panels depict trajectories of individual growth cone migrations. The initial growth cone positions are normalized at (x = 0 μm, y = 0 μm). See also quantitative data in *Figure 7—figure supplement 3*. Bars: 50 μm. Data represent means ± SEM; ***p<0.01 (one-way ANOVA with Schaffer's post hoc test).
DOI: https://doi.org/10.7554/eLife.34593.034

The following source data and figure supplements are available for figure 7:

**Source data 1.** Quantitative data for axon outgrowth velocity related to *Figure 7C*.
DOI: https://doi.org/10.7554/eLife.34593.048
**Source data 2.** Quantitative data for turning angle of axon toward the netrin-1 source related to *Figure 7D*.
DOI: https://doi.org/10.7554/eLife.34593.049
**Figure supplement 1.** Soluble and substrate-bound netrin-1 contribute to axon turning.
DOI: https://doi.org/10.7554/eLife.34593.035
**Figure supplement 1—source data 1.** Quantitative data for axon outgrowth velocity related to *Figure 7—figure supplement 1B*.
DOI: https://doi.org/10.7554/eLife.34593.036
**Figure supplement 1—source data 2.** Quantitative data for turning angle of axon toward the netrin-1 source related to *Figure 7—figure supplement 1C*.
DOI: https://doi.org/10.7554/eLife.34593.037
**Figure supplement 2.** *Shootin1* knockout leads to inhibition of netrin-1–induced axon outgrowth and turning.
DOI: https://doi.org/10.7554/eLife.34593.038
**Figure supplement 2—source data 1.** Quantitative data for axon outgrowth velocity related to *Figure 7—figure supplement 2C*.
DOI: https://doi.org/10.7554/eLife.34593.039
**Figure supplement 2—source data 2.** Quantitative data for turning angle of axon toward the netrin-1 source related to *Figure 7—figure supplement 2D*.
DOI: https://doi.org/10.7554/eLife.34593.040
**Figure supplement 3.** Shootin1a–L1-CAM interaction mediates netrin-1–induced axon guidance.
DOI: https://doi.org/10.7554/eLife.34593.041
**Figure supplement 3—source data 1.** Quantitative data for axon outgrowth velocity related to *Figure 7—figure supplement 3A*.
DOI: https://doi.org/10.7554/eLife.34593.042
**Figure supplement 3—source data 2.** Quantitative data for turning angle of axon toward the netrin-1 source related to *Figure 7—figure supplement 3B*.
DOI: https://doi.org/10.7554/eLife.34593.043
**Figure supplement 4.** Shootin1a–L1-CAM interaction mediates netrin-1–induced axon guidance on laminin.
DOI: https://doi.org/10.7554/eLife.34593.044
**Figure supplement 4—source data 1.** Quantitative data for axon outgrowth velocity related to *Figure 7—figure supplement 4B*.
DOI: https://doi.org/10.7554/eLife.34593.045
**Figure supplement 4—source data 2.** Quantitative data for turning angle of axon toward the netrin-1 source related to *Figure 7—figure supplement 4C*.
DOI: https://doi.org/10.7554/eLife.34593.046
**Figure supplement 4—source data 3.** Quantitative data for F-actin retrograde flow speed related to *Figure 7—figure supplement 4D*.
DOI: https://doi.org/10.7554/eLife.34593.047

## Discussion

Since the seminal proposal by Ramón y Cajal (*Cajal, 1890*) that the growth cone senses extracellular chemical cues and produces force for axon guidance, considerable progress has been made in understanding the signaling events at the axon guidance machinery located within the growth cone. However, the gradient-reading as well as mechano-effector machinery that converts the environmental spatial chemical signals into the directional force for axon guidance has remained unclear. Here we have shown that shallow gradients of netrin-1 elicited highly polarized shootin1a phosphorylation within growth cones. Netrin-1–elicited shootin1a phosphorylation promoted direct interaction between shootin1a and L1-CAM, thereby generating traction force for growth cone motility.

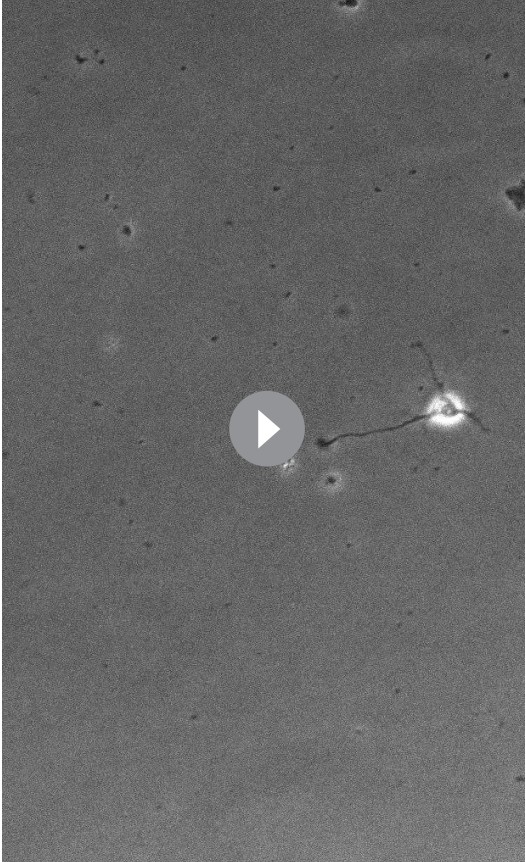

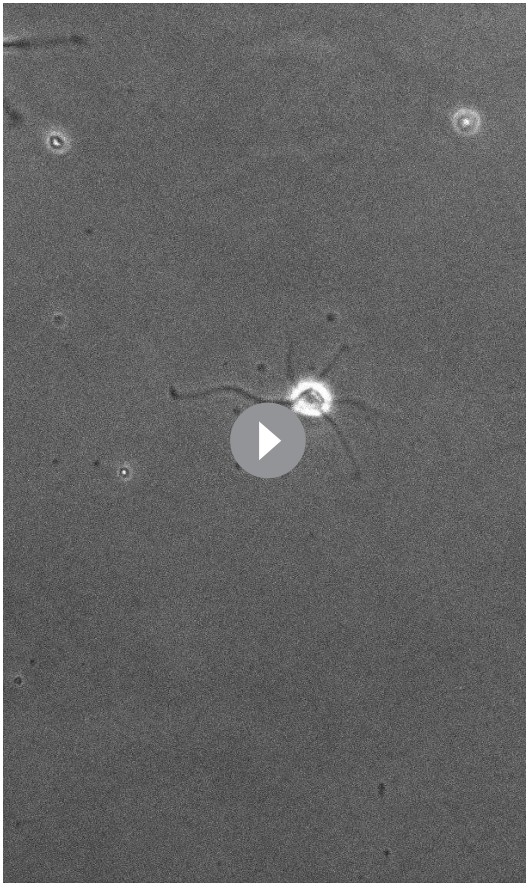

**Video 6.** A time-lapse movie of a hippocampal neuron expressing control miRNA, under a gradient of netrin-1. See the legend for *Figure 7A*. The gradient of Alexa Fluor 594-BSA in *Figure 7A* is not shown.
DOI: https://doi.org/10.7554/eLife.34593.050

**Video 7.** A time-lapse movie of a hippocampal neuron expressing shootin1a miRNA, under a gradient of netrin-1. See the legend for *Figure 7B*. The gradient of Alexa Fluor 594-BSA in *Figure 7B* is not shown.
DOI: https://doi.org/10.7554/eLife.34593.051

Furthermore, the spatially regulated phosphorylation of shootin1a within growth cones was required for axon turning induced by netrin-1 gradients. These results suggest that shootin1a constitutes a gradient-reading and mechano-effector machinery involved in netrin-1–induced axon guidance.

## Mechano-effector machinery for netrin-1–regulated axon guidance

The present study defines a mechano-effector for netrin-1–induced axon guidance. Previous reports proposed that an increase in the mechanical coupling between F-actin retrograde flow in the growth cone and cell adhesions transmits the force of F-actin flow onto the adhesive substrates for growth cone migration (*Mitchison and Kirschner, 1988*; *Suter and Forscher, 2000*). In relation to this notion, a recent study reported that axon guidance cues affect F-actin–adhesion coupling locally within the growth cone to influence axon outgrowth and guidance (*Nichol et al., 2016*). In addition, we previously reported that shootin1a associates with the F-actin flow at the growth cone through its direct interaction with cortactin, and showed that netrin-1–induced phosphorylation of shootin1a by Pak1 promotes shootin1a–F-actin interaction through cortactin (*Toriyama et al., 2013*; *Kubo et al., 2015*).

Here, we demonstrated that netrin-1–induced shootin1a phosphorylation by Pak1 also promotes direct interaction between shootin1a and L1-CAM. Previous studies reported that netrin-1 induces dimerization of its receptor, DCC, thereby activating molecules including NCK1, FAK and FYN (*Stein et al., 2001*; *Ren et al., 2004*; *Lai Wing Sun et al., 2011*). This in turn induces activation of

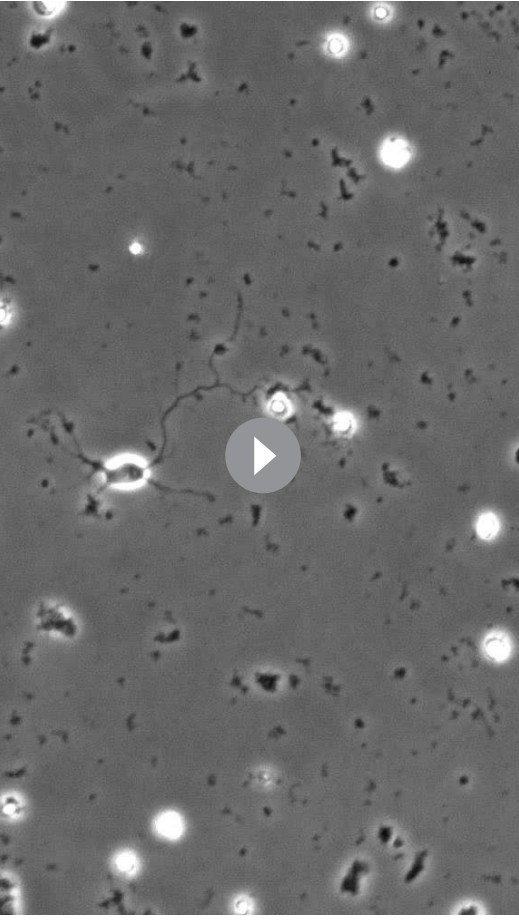

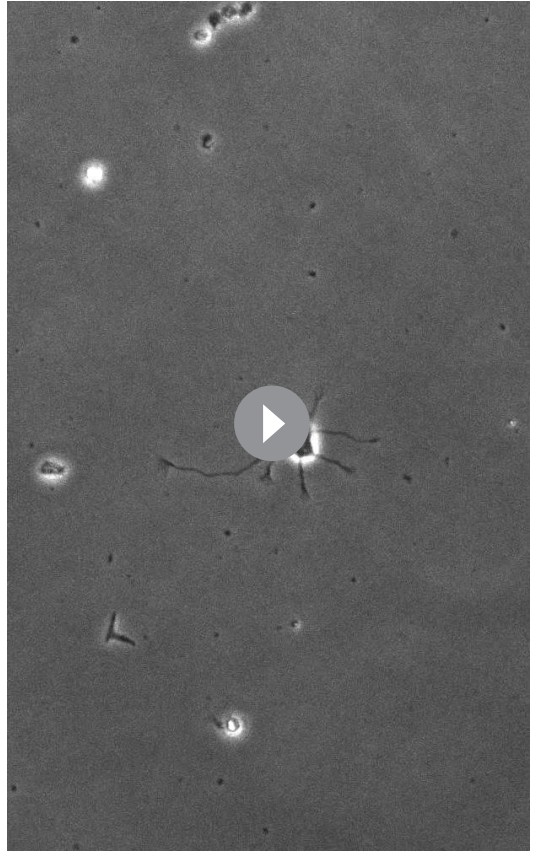

**Video 8.** A time-lapse movie of a hippocampal neuron expressing EGFP, under a gradient of netrin-1. See the legend for *Figure 7E*. The gradient of Alexa Fluor 594-BSA in *Figure 7E* is not shown.
DOI: https://doi.org/10.7554/eLife.34593.052

**Video 9.** A time-lapse movie of a hippocampal neuron expressing EGFP-shootin1a (1-125), under a gradient of netrin-1. See the legend for *Figure 7F*. The gradient of Alexa Fluor 594-BSA in *Figure 7F* is not shown.
DOI: https://doi.org/10.7554/eLife.34593.053

Cdc42 and Rac1, and their downstream kinase Pak1 (*Li et al., 2002*; *Shekarabi and Kennedy, 2002*; *Shekarabi et al., 2005*; *Briançon-Marjollet et al., 2008*; *Demarco et al., 2012*) (*Figure 8E*). Thus, under the activation of Pak1 via these signaling pathways, the shootin1a phosphorylation enhances both the shootin1a–adhesion and shootin1a–F-actin interactions that lead to increased F-actin–adhesion coupling (*Figure 8E*). This double regulation would enable efficient regulation of forces for axon guidance in response to netrin-1. At present, no information is available on the three-dimensional structure of shootin1a. In addition, it is unknown how shootin1a–L1-CAM and shootin1a–cortactin interactions are promoted by the phosphorylation of shootin1a. Future investigations of the molecular structure of shootin1a, and of how the structures of the domains mediating interaction with L1-CAM and cortactin are affected by phosphorylation, will lead to a better understanding of this mechano-effector machinery.

## Gradient-reading machinery for netrin-1–regulated axon guidance

The ability of cells to sense small spatial differences in environmental cues is essential for proper axon guidance as well as directional cell migration, but the molecular mechanism underlying it remains a major question (*Quinn and Wadsworth, 2008*; *Hegemann and Peter, 2017*). Initial studies reported that growth cones can respond to 1% gradients of repulsive tectal membranes presented on the substrate (*Baier and Bonhoeffer, 1992*) and 5–10% gradients of diffusible axon guidance molecules including netrin-1 (*Ming et al., 1999*). More recently, analyses employing microfluidic devices demonstrated that growth cones have even higher sensitivities to chemical gradients:

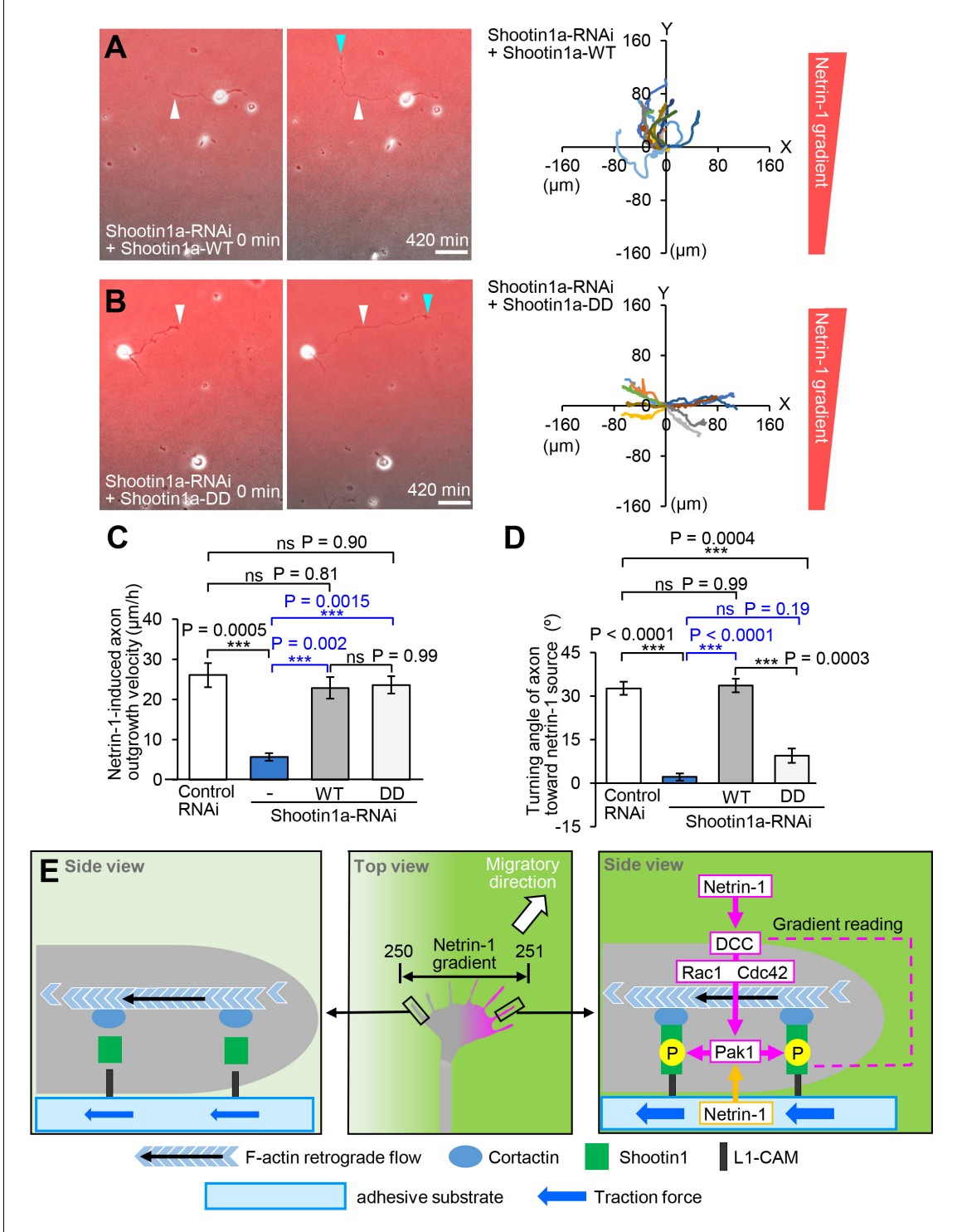

**Figure 8.** Asymmetric shootin1a phosphorylation within growth cones is required for netrin-1–induced axon guidance. (**A** and **B**) Time-lapse phase-contrast/fluorescence images of hippocampal neurons expressing shootin1a miRNA + RNAi refractory shootin1a-WT (**A**), and shootin1a miRNA + RNAi refractory shootin1a-DD (**B**) under gradients of netrin-1 and Alexa Fluor 594-BSA (red). White and blue arrowheads indicate growth cones at the first and last time-points, respectively. See *Videos 10* and *11*. The right panels depict trajectories of individual growth cone migrations. The initial growth cone positions are normalized at (x = 0 μm, y = 0 μm). (**C**) Axon outgrowth velocity obtained from the analyses in *Figure 7A and B*, *Figure 8A and B* (*n* = 47 growth cones). (**D**) Turning angle of axon toward the netrin-1 source was obtained from the analyses in *Figure 7A and B*, *Figure 8A and B* (*n* = 47 growth cones), by calculating the difference between the angles of the axonal tip at the first and last time-points of the observations (*θ*f - *θ*l). (**E**) A model for gradient-reading and mechanoresponse processes of netrin1–induced axon guidance. A very small difference (250:251; 0.4%) in netrin-1

*Figure 8 continued on next page*

*Figure 8 continued*

concentration can induce highly polarized phosphorylation of shootin1a within growth cones (pink), as a readout of highly sensitive gradient-reading processes. A netrin-1 gradient on the substrate would also contribute to polarized shootin1 phosphorylation (yellow). This process is achieved through a signaling pathway including DCC, Rac1/CDC42, Pak1 and shootin1a. The polarized phosphorylation of shootin1a within a growth cone locally promotes shootin1a–L1-CAM and shootin1a–cortactin interactions. These interactions in turn enhance asymmetrically the coupling between F-actin retrograde flow and the adhesive substrate and increase traction force (blue arrows) on the side of the netrin-1 source, thereby leading to a decision for the migratory direction (white arrow). Data represent means ± SEM; ***$p < 0.01$; ns, not significant (one-way ANOVA with Schaffer's post hoc test). Bars: 50 µm.

DOI: https://doi.org/10.7554/eLife.34593.054

The following source data is available for figure 8:

**Source data 1.** Quantitative data for axon outgrowth velocity related to *Figure 8C*.
DOI: https://doi.org/10.7554/eLife.34593.055
**Source data 2.** Quantitative data for turning angle of axon toward the netrin-1 source related to *Figure 8D*.
DOI: https://doi.org/10.7554/eLife.34593.056

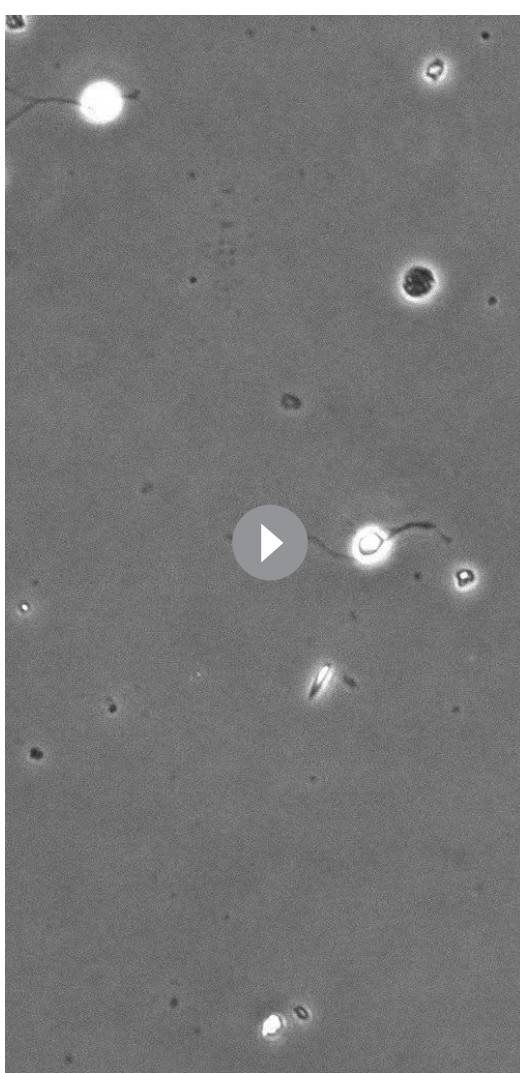

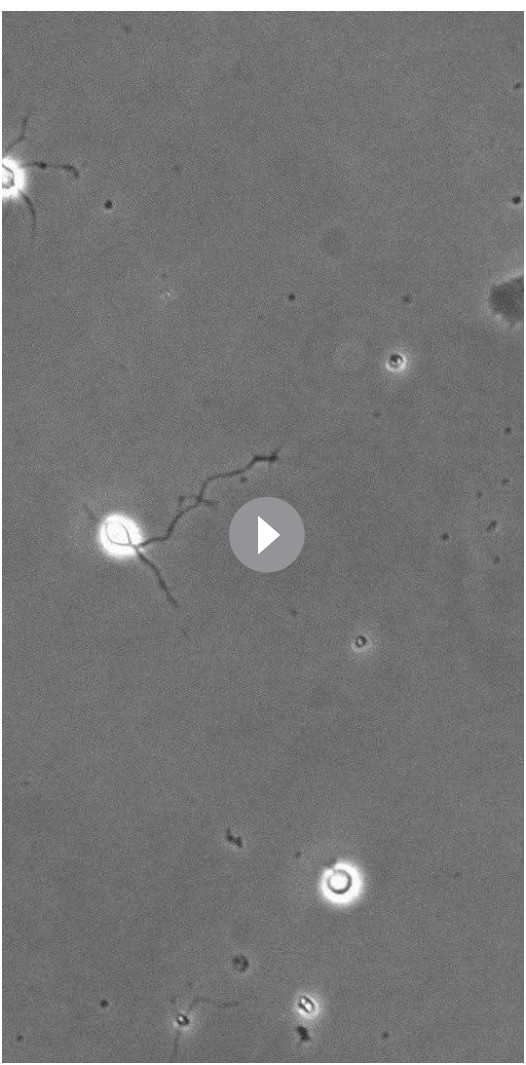

**Video 10.** A time-lapse movie of a hippocampal neuron expressing shootin1a miRNA + RNAi refractory shootin1a-WT, under a gradient of netrin-1. See the legend for *Figure 8A*. The gradient of Alexa Fluor 594-BSA in *Figure 8A* is not shown.
DOI: https://doi.org/10.7554/eLife.34593.057

**Video 11.** A time-lapse movie of a hippocampal neuron expressing shootin1a miRNA + RNAi refractory shootin1a-DD, under a gradient of netrin-1. See the legend for *Figure 8B*. The gradient of Alexa Fluor 594-BSA in *Figure 8B* is not shown.
DOI: https://doi.org/10.7554/eLife.34593.058

for example, they can turn in response to 0.1–0.4% gradients of NGF (*Rosoff et al., 2004*), and 0.5 and 0.1% gradients of substrate-bound laminin and ephrin-A5, respectively (*Xiao et al., 2014*). This study presents the framework of a highly sensitive gradient-reading machinery for axon guidance (*Figure 8E*). Our data demonstrate that a 0.4% (250:251) difference in netrin-1 concentration induces a 71% difference in shootin1a phosphorylation within growth cones, as a key readout of the spatial signal. This polarized phosphorylation locally promotes shootin1a–L1-CAM and shootin1a–cortactin interactions within growth cones and asymmetrically promote traction force (blue arrows, F *Figure 8E*) on the side of the netrin-1 source, leading to a decision for the migratory direction (white arrow, *Figure 8E*).

Our data suggest that the netrin-1 gradient-reading process in growth cones is achieved through a signaling pathway that includes DCC, Rac1/Cdc42, Pak1 and shootin1a (*Figure 8E*); however, how these molecules amplify very small spatial differences in netrin-1 concentration remain to be determined. It has been proposed that amplification of local signals through combined positive and negative feedback loops contribute to sense shallow gradients of extracellular chemicals (*Yang et al., 2016*; *Hegemann and Peter, 2017*) and that polarized assembly of signaling molecules may play a key role in it (*Quinn and Wadsworth, 2008*; *Hegemann and Peter, 2017*). Previous studies in *C. elegans* reported that local netrin-1 signals induce polarized distribution of the DCC orthologs UNC-40 within cell bodies (*Adler et al., 2006*; *Ziel et al., 2009*; *Wang et al., 2014*). However, growth cones of mouse cortical neurons did not show polarized localization of DCC under netrin-1 gradients (*Taylor et al., 2015*). In addition, FRET visualization of Cdc42 and Rac1 signals has not, so far, revealed a distinct polarized activation of these molecules in growth cones (*Picard et al., 2009*; *Rappaz et al., 2016*). Tracing the spatial signals from phosphorylated shootin1a back upstream to DCC will lead to a better molecular understanding of the gradient-reading mechanism involved in netrin-1–regulated axon guidance.

## Shootin1a–mediated axon guidance in the brain

The present study has shown that shootin1a is expressed at high levels in developing forebrain commissural axons and that *Shootin1* knockout mice display dysgenesis and misprojection of these axons. We analyzed their projections with the axonal marker L1-CAM (*Chung et al., 1991*; *Klingler et al., 2015*) and DiI tracing (*Klingler et al., 2015*). As L1-CAM interacts with shootin1a, ablation of shootin1a expression could lead to a change in L1-CAM localization in these axons; however, both the L1-CAM labelling and DiI tracing analyses revealed misprojection of these axons. Similar defects of callosal and anterior commissural axons were reported in knockout mice for *Netrin-1* (*Serafini et al., 1996*) as well as *DCC* (*Fazeli et al., 1997*), *Rac1* (*Chen et al., 2007*; *Kassai et al., 2008*) and *L1-CAM* (*Demyanenko et al., 1999*), thereby providing evidence for the notion that shootin1a cooperates with these molecules in axon guidance. Thus, the defects in the forebrain commissural axons in *Shootin1* knockout mice are consistent with the in vitro observations that shootin1a is required for netrin-1–induced axon outgrowth and guidance.

However, although netrin-1 mRNA is distributed along the paths of forebrain commissural axons (*Serafini et al., 1996*), netrin-1 gradients have not yet been reported in these brain regions. In addition, as *Shootin1* knockout mice exhibit multiple defects in the brain, we cannot conclude that the dysgenesis of the forebrain commissures is only due to the axon outgrowth and guidance deficits observed in in vitro assays. Generation of conditional knockout mice as well as detailed mapping of netrin-1 will facilitate future analyses of the shootin1a–mediated axon guidance in the brain. In contrast to the forebrain commissural axons, we could not detect distinct shootin1a localization in the ventral spinal commissural axons and could not observe their abnormality in *Shootin1* knockout mice, suggesting that shootin1a is not required for guidance of the spinal commissural axons.

## Netrin-1–induced axon guidance through chemotaxis and haptotaxis

Axon guidance and cell migration are directed by spatial gradients of soluble chemicals (called chemotaxis) (*Gundersen and Barrett, 1979*; *Mortimer et al., 2008*) and by chemicals presented on adhesive substrate or neighboring cells (termed haptotaxis) (*Carter, 1967*; *Baier and Bonhoeffer, 1992*). The present study indicates that gradients of both soluble and immobilized netrin-1 are produced on L1-CAM–coated substrate, thereby contributing to axon turning of hippocampal neurons. As soluble and immobilized chemical cues act as ligands to activate intracellular signaling pathways

in growth cones (*Huber et al., 2003*; *Lowery and Van Vactor, 2009*; *Kolodkin and Tessier-Lavigne, 2011*), the gradient-reading machinery involving shootin1a may explain both axonal chemotaxis and haptotaxis elicited by shallow gradients of netrin-1 (red and yellow arrows, *Figure 8E*).

On the other hand, recent studies reported that netrin-1 locally presented by neural progenitors is required for guidance of the spinal commissural axons (*Dominici et al., 2017*; *Varadarajan et al., 2017*) and that these axons respond to substrate-bound netrin-1 (*Moore et al., 2012*) and steep gradients of netrin-1 (*Sloan et al., 2015*) in vitro. These reports underscore the importance of haptotaxis in netrin-1–induced axon guidance of spinal commissural neurons. Concerning the axon guidance mechanism of these neurons, *Moore et al. (2012)* reported that immobilization of netrin-1 is required; the immobilized netrin-1 is proposed to play a key role in mechanical activation of FAK, thereby leading to activation of signaling pathways including Crk-associated substrate (CAS) for axonal haptotaxis. The details of how axon guidance is regulated by immobilized netrin-1 remain important issues for future analyses.

# Materials and methods

## Key resources table

| Reagent type (species) or resource | Designation | Source or reference | Identifiers | Additional information |
|---|---|---|---|---|
| Gene (*Rattus norvegicus*) | Wistar | SLC | RRID:RGD_2314928 | |
| Gene (*Rattus norvegicus*) | Wistar | CLEA Japan | RRID:RGD_12879431 | |
| Gene (*Mus musculus*) | C57BL/6 | SLC | RRID:MGI:5658686; RRID:MGI:5295404 | |
| Gene (*Mus musculus*) | C57BL/6 | CLEA Japan | RRID:MGI:5658686; RRID:MGI:2160139 | |
| Genetic reagent (*Mus musculus*) | *Shootin1* gene knockout | This paper | | Please see 'Generation of *Shootin1* knockout mice' in Materials and methods section |
| Cell line (*Homo sapiens*) | HEK293T cell | ATCC | Cat# CRL_3216; RRID:CVCL_0063 | |
| Antibody | anti-shootin1a peptide sequence (rabbit polyclonal) | This paper | | Rabbit polyclonal; against aa 450–456; Immunohistochemistry: (1:5000) Immunoblot: (1:5000) |
| Antibody | anti-shootin1 antibody (rabbit polyclonal) | PMID: 17030985 (*Toriyama et al., 2006*); PMID: 23453953 (*Toriyama et al., 2013*) | | Immunoblot: (1:1000) |
| Antibody | anti-pSer101-shootin1 antibody (rabbit polyclonal) | PMID: 23453953 (*Toriyama et al., 2013*) | | Immunoblot: (1:1000) |
| Antibody | anti-pSer249-shootin1 antibody (rabbit polyclonal) | PMID: 23453953 (*Toriyama et al., 2013*) | | Immunoblot: (1:5000); Immunofluorescence(1:1000) |
| Antibody | anti-NCAM-L1 (C-20) antibody (goat polyclonal) | Santa Cruz Biotechnology | Cat# sc-1508; RRID:AB_631086 | Immunoblot: (1:2000); Immunofluorescence: (1:1000); Immunohistochemistry: (1:1000) |
| Antibody | anti-Neurofilament antibody 2H3 (mouse monoclonal) | DSHB | Cat# 2H3; RRID:AB_531793 | Immunohistochemistry: (1:2000) |
| Antibody | anti-TAG-1 antibody 4D7 (mouse monoclonal) | DSHB | Cat# 4D7/TAG1; RRID:AB_531775 | Immunohistochemistry: (1:100) |
| Antibody | anti-FLAG(DDDDK) tag antibody (rabbit polyclonal) | MBL | Cat# PM020; RRID:AB_591224 | Immunoblot: (1:1000) |

*Continued on next page*

*Continued*

| Reagent type (species) or resource | Designation | Source or reference | Identifiers | Additional information |
|---|---|---|---|---|
| Antibody | anti-Myc tag antibody (rabbit polyclonal) | MBL | Cat# 562–5; RRID:AB_591116 | Immunoblot: (1:2000) |
| Antibody | anti-GST tag antibody (goat polyclonal) | GE Healthcare | Cat# 27-4577-01; RRID:AB_771432 | Immunoblot: (1:3000) |
| Antibody | anti-His tag antibody (goat polyclonal) | Wako | Cat# 014–23221 | Immunofluorescence (1:500) |
| Antibody | anti-rabbit IgG secondary antibody, Alexa Fluor 594 (from donkey) | Jackson immune research | Cat# 711-585-152; RRID:AB_2340621 | Immunofluorescence: (1:1000); Immunohistochemistry: (1:1000) |
| Antibody | anti-goat IgG secondary antibody, Alexa Fluor 488 (from donkey) | Invitrogen, Thermo Fisher Scientific | Cat# A-11055; RRID:AB_2534102 | Immunofluorescence: (1:1000); Immunohistochemistry: (1:1000) |
| Antibody | anti-mouse IgG secondary antibody, Alexa Fluor 488 (from goat) | Invitrogen, Thermo Fisher Scientific | Cat# A-11029; RRID:AB_2534088 | Immunofluorescence: (1:1000) |
| Antibody | anti-rabbit IgG, Whole Ab ECL antibody, HRP Conjugated (from donkey) | GE Healthcare | Cat# NA934; RRID:AB_772206 | Immunoblot: (1:2000) |
| Antibody | anti-goat IgG, HRP conjugate, Species Adsorbed: H, M, R, Ch, Gp, Eq, Ht, Rb antibody (from donkey) | Millipore | Cat# AP180P; RRID:AB_92573 | Immunoblot: (1:2000) |
| Recombinant DNA reagent | pCMV-myc vector | Stratagene, Agilent | Cat# 211173 | |
| Recombinant DNA reagent | pCMV-Flag vector | Strategene, Agilent | Cat# 211172 | |
| Recombinant DNA reagent | pCAGGS vector | PMID: 1660837 (*Niwa et al., 1991*) Strategene, Addgene | Collection number LMBP2453 | This vector was provided by J. Miyazaki, Osaka University, Osaka, Japan; *Niwa et al. (1991)* (PMID: 1660837) |
| Recombinant DNA reagent | pGEX-6P-1 | GE Healthcare | Cat# 28954648 | |
| Recombinant DNA reagent | pCMV-mRFP-actin vector | PMID: 29483251 (*Abe et al., 2018*) | | |
| Recombinant DNA reagent | pCMV-myc-shootin1a vector | PMID: 17030985 (*Toriyama et al., 2006*) | | |
| Recombinant DNA reagent | pCMV-Flag-L1-CAM-ICD (intracellular domain) vector | This paper | | |
| Recombinant DNA reagent | pCMV-dominant negative myc-Pak1vector | PMID: 26261183 (*Kubo et al., 2015*) | | |
| Recombinant DNA reagent | pCMV-constitutively active myc-Pak1 vector | PMID: 26261183 (*Kubo et al., 2015*) | | |
| Recombinant DNA reagent | pCAGGS-myc | PMID: 17030985 (*Toriyama et al., 2006*) | | |
| Recombinant DNA reagent | pCAGGS-myc-GST vector | PMID: 17030985 (*Toriyama et al., 2006*) | | |
| Recombinant DNA reagent | pCAGGS-myc- shootin1a (1-125) vector | This paper | | |
| Recombinant DNA reagent | pCAGGS-EGFP | This paper | | |
| Recombinant DNA reagent | pCAGGS-EGFP- shootin1a(1-125) | This paper | | |
| Recombinant DNA reagent | pGEX-shootin1a-WT vector | PMID: 26261183 (*Kubo et al., 2015*) | | |

*Continued on next page*

*Continued*

| Reagent type (species) or resource | Designation | Source or reference | Identifiers | Additional information |
|---|---|---|---|---|
| Recombinant DNA reagent | pGEX-shootin1a-DD (phopho-mimic shootin1a) vector | PMID: 26261183 (*Kubo et al., 2015*) | | |
| Recombinant DNA reagent | pGEX-myc-shootin1a-DD (phopho-mimic shootin1a) vector | PMID: 26261183 (*Kubo et al., 2015*) | | |
| Recombinant DNA reagent | pGEX-myc-shootin1a-(1-125) vector | PMID: 26261183 (*Kubo et al., 2015*) | | |
| Recombinant DNA reagent | pGEX-myc-shootin1a-(125-260) vector | PMID: 26261183 (*Kubo et al., 2015*) | | |
| Recombinant DNA reagent | pGEX-myc-shootin1a-(217-456) vector | PMID: 26261183 (*Kubo et al., 2015*) | | |
| Recombinant DNA reagent | pGEX-myc-shootin1a-(261-377) vector | PMID: 26261183 (*Kubo et al., 2015*) | | |
| Recombinant DNA reagent | pGEX-L1-CAM-ICD (intracellulardomain) vector | This paper | | |
| Recombinant DNA reagent | RNAi shootin1a vector (miRNA) | PMID: 17030985 (*Toriyama et al., 2006*); PMID: 23453953 (*Toriyama et al., 2013*) | | |
| Recombinant DNA reagent | RNAi-refractory shootin1a-WT vector | PMID: 23453953 (*Toriyama et al., 2013*) | | |
| Recombinant DNA reagent | RNAi-refractory shootin1a-DD (phopho-mimic shootin1a) vector | PMID: 23453953 (*Toriyama et al., 2013*) | | |
| Peptide, recombinant protein | recombinant Netrin-1 protein (from mouse) | R and D systems | Cat# 1109-N1-025 | No CF (No carrier protein free) |
| Peptide, recombinant protein | FLAG peptide | Sigma-Aldrich | Cat# F3290 | |
| Peptide, recombinant protein | Laminin Solution, from Mouse EHS Tumor | Wako | Cat# 120–05751 | |
| Peptide, recombinant protein | L1-CAM-Fc | PMID: 18519736 (*Shimada et al., 2008*) | N/A | |
| Peptide, recombinant protein | Prescission protease | GE Healthcare | Cat# 27084301 | |
| Peptide, recombinant protein | Recombinant Pak1 | Life technologies, Thermo Fisher Scientific | Cat# PV3820 | |
| Commercial assay or kit | Rat Neuron Nucleofector kits (25 RCT) | Lonza | Cat# VPG-1003 | |
| Chemical compound, drug | anti-FLAG M2 antibody affinity gel (mouse monoclonal) | Sigma-Aldrich | Cat# A2220; RRID:AB_10063035 | |
| Chemical compound, drug | 7-amino-4-chloromethylcoumarin (CMAC) | Invitrogen, Thermo Fisher Scientific | Cat# C2110 | |
| Chemical compound, drug | 4,6-diamidino-2-phenylindole(DAPI) | Roche | Cat# 10236276001 Roche | DAPI stain (1:1000) |
| Chemical compound, drug | 1,1'-dioctadecyl-3,3,3',3'-tetramethyl-indocarbocyanine dye (DiI) | Invitrogen, Thermo Fisher Scientific | Cat# D3911 | |
| Chemical compound, drug | Glutathione sepharose 4B | GE Healthcare | Cat# 17-0756-01 | |

*Continued on next page*

*Continued*

| Reagent type (species) or resource | Designation | Source or reference | Identifiers | Additional information |
|---|---|---|---|---|
| Chemical compound, drug | Protein G-sepharose 4B | GE Healthcare | Cat# 6511–5 | |
| Chemical compound, drug | Polydimethylsiloxane (PDMS) | Dow Corning Toray, Japan | Cat# 3255981 | |
| Chemical compound, drug | Silicone oil (Barrier coat No.6) | ShinEtsu, Japan | Cat# 06003 | |
| Chemical compound, drug | PhosSTOP | Roche | Cat# 4906845001 | |
| Software, algorithm | Image J | https://imagej.nih.gov/ij/ | RRID:SCR_003070 | |
| Software, algorithm | Fiji | http://fiji.sc | RRID:SCR_002285 | |
| Software, algorithm | Graphpad prism 7 | Graphpad software | RRID:SCR_002798 | |
| Software, algorithm | R Project for Statistical Computing | http://www.r-project.org/ | RRID:SCR_001905 | |
| Software, algorithm | Matlab | http://www.mathworks.com/products/matlab/ | RRID:SCR_001622 | |
| Software, algorithm | Microsoft Excel 2016 | Microsoft https://www.microsoft.com | | |
| Other | Amicon ultra-4 centrifugal filter devices | Millipore | Cat# UFC800324 | |

## Histology and immunohistochemistry

All relevant aspects of the experimental procedures were approved by the Institutional Animal Care and Use Committee of Nara Institute of Science and Technology (reference No. 1802). For timed pregnancy, the morning of vaginal plug detection was designated as embryonic day E0.5. The brains (E16.5 and P0) and embryos (E12.5) were fixed by immersion in 4% formaldehyde (FA) prepared fresh from paraformaldehyde (PFA) at 4°C for 60 min. Serial sections (8 μm) of paraffin-embedded brains were cut on a microtome (Micro-edge Instruments) and used for Nissl substance staining. For immunohistochemistry, 12 μm cryosections cut by a cryostat (Leica) were preincubated with 10% fetal bovine serum (Invitrogen) in 1 × phosphate buffer (PB) containing 0.3% Triton-X 100 for 2 hr. The sections were then incubated with the primary antibodies at 4°C two overnight; the primary antibodies used were rabbit anti-shootin1a (1:5,000), goat anti-L1-CAM (Santa Cruz, RRID:AB_631086) (1:1,000), mouse anti-Neurofilament (Cat# 2H3, RRID:AB_531793) (1:2,000), and mouse anti-TAG-1 (Cat# 4D7/TAG1, RRID:AB_531775) (1:100) diluted in PB containing 0.3% Triton-X 100. Secondary antibodies were Alexa Fluor 488 anti-goat (Invitrogen, RRID:AB_2534102) and Alexa Fluor 594 anti-rabbit (Invitrogen, RRID:AB_2340621): they were used at a 1000-fold dilution at 4°C overnight. The WT and knockout sections used were from the same coronal or horizontal stereotaxic brain regions. Slides were mounted in 50% glycerol (v/v) in PBS after staining with 4,6-diamidino-2-phenylindole (DAPI; Roche). Fluorescence images were acquired using a confocal microscope (LSM 700 or LSM 710; Carl Zeiss) equipped with a plan-Apochromat ×10, 0.45 NA and ×20, 0.8 NA objective lens (Carl Zeiss) or BZ-X700 fluorescence microscope (Keyence) equipped with a CFI Plan Apo ×10, 0.45 NA objective lens (Nikon).

## Generation of *Shootin1* knockout mice

The targeting vector for *Shootin1* knockout mice was constructed to replace most of the first exon of *Shootin1* with IRES-LacZ and PGK-neo (*Figure 1—figure supplement 3A*). The linearized targeting vector was introduced into 129 donor ES cells and the mutation of *Shootin1* in the cells was confirmed by Southern blot analysis. Targeted ES clones were microinjected into C57BL/6 blastocysts and implanted into pseudopregnant mice. Chimeric mice were crossed with C57BL/6 mice for at least seven generations before analysis.

## Dil tracing

1,1'-dioctadecyl-3,3,3',3'-tetramethyl-indocarbocyanine dye (DiI; Invitrogen) crystals were placed in the dorsomedial cortex in a rostrocaudal series for tracing of the corpus callosum, or in the anterior olfactory nucleus/anterior piriform cortex for tracing of the anterior limb of the anterior commissure (*Klingler et al., 2015*). After the placement, brains were incubated in 4% FA at 37°C for 3 months, and then 100 μm sections were prepared by vibratome (Leica). The sections were mounted on glass slides, and observed under a fluorescence microscope.

## Cultures and transfection

Hippocampal neurons prepared from E18 rats were cultured on glass coverslips coated sequentially with polylysine and L1-CAM-Fc as described (*Shimada et al., 2008*; *Toriyama et al., 2013*; *Kubo et al., 2015*). For the experiments in *Figure 7—figure supplement 4*, we cultured neurons on glass coverslips coated sequentially with polylysine and laminin as described (*Toriyama et al., 2006*; *Abe et al., 2018*). All experiments except for the measurement of forces were carried out on glass surfaces. For the immunoprecipitation and immunoblot analyses in *Figure 4C*, we used cortical neurons, which also respond to netrin-1 (*Li et al., 2008*), as the experiments required large numbers of neurons. They were prepared from E18 rat embryos using the same protocol as above. The neurons were transfected with vectors using Nucleofector (Lonza) before plating. HEK293T cells (ATCC, RRID:CVCL_0063, cell identities were authenticated by STR profiling and cells were tested negative for mycoplasma using the TAKARA PCR mycoplasma detection set Cat# 6601) were cultured in Dulbecco's modified Eagle's medium supplemented with 10% fetal bovine serum and transfected with plasmid DNA by the calcium phosphate method.

## Axon guidance assay

A microfluidic device that generates netrin-1 gradients in culture medium was produced according to a previous report (*Bhattacharjee et al., 2010*), with modification. Briefly, it was fabricated with polydimethylsiloxane (PDMS; Silpot 184, Dow Corning Toray, Japan) and a glass coverslip; the device consists of an open rectangular cell culture area and two microchannels on the long sides of the culture area (*Figure 2A*). The micro-molds of the channel pattern were lithographically fabricated on a photoresist (SU-8 3025, MicroChem, USA) spin-coated on a 70 μm thick silicon wafer. PDMS sheets were obtained from this mold, which had been treated with silicone oil (Barrier coat No. 6, ShinEtsu, Japan) to facilitate their removal. A PDMS sheet, coated with 1 μm thick PDMS glue (KE103, ShinEtsu, Japan), was then bonded to a glass coverslip coated sequentially with polylysine and L1-CAM-Fc. To generate netrin-1 gradients in the cell culture area, flows of culture medium (7.5 μm/min) with or without 4.4 nM (300 ng/ml) netrin-1 and 2 μM fluorescent tracer (Alexa Fluor 594-BSA or Alexa Fluor 488-BSA) were applied to the microchannels on either side of the open cell culture area (black arrows, *Figure 2A*). As reported (*Bhattacharjee et al., 2010*), the microfluidic device generated stable gradients of the tracer in the culture medium (*Figure 2B*). For live imaging of neurons expressing miRNA, EGFP fluorescence was used as an indicator of miRNA expression. The turning angle of an axon toward the netrin-1 source was obtained by calculating the difference between the angles of the axonal tip at the first and last time-points of the observations ($\theta$f - $\theta$l) (*Bhattacharjee et al., 2010*) (*Figure 7D*).

## RNAi

For RNAi experiments, we used a Block-iT Pol II miR RNAi expression kit (Invitrogen). The targeting sequence of shootin1a miRNA and its effectiveness were reported previously (*Toriyama et al., 2006*). As described previously (*Shimada et al., 2008*), to ensure high-level expression of miRNA before neurite elongation, hippocampal neurons prepared from E18 rat embryos and transfected with the miRNA expression vector were plated on uncoated polystyrene plates. After a 20 hr incubation to induce miRNA expression, the cells were collected and then cultured on coverslips.

## DNA constructs

Preparation of the vectors to express shootin1a-WT has been described previously (*Toriyama et al., 2006*). cDNA fragments of shootin1a deletion mutants were amplified by PCR and subcloned into pGEX-6P-1 (GE Healthcare), pCAGGS-myc, pCAGGS-EGFP, pCMV-myc (Stratagene) or pEGFP

(Clontech) vector as described (*Kubo et al., 2015*). The generation of RNAi-refractory shootin1a-WT and shootin1a-DD shootin1a was described previously (*Toriyama et al., 2013*). pCAGGS-myc was used to overexpress proteins under the β-actin promoter as described (*Toriyama et al., 2006*).

## Protein preparation and in vitro kinase assay

Recombinant proteins were expressed in *Escherichia coli* as GST fusion proteins and purified on Glutathione sepharose columns, after which GST was removed by PreScission protease. L1-CAM-Fc was prepared as described (*Shimada et al., 2008*). Kinase reactions were carried out in 20 µl kinase buffer (50 mM HEPES pH 7.5, 10 mM MgCl$_2$, 2 mM MnCl$_2$, 1 mM DTT, 125 µM ATP, in the presence or absence of 10 µCi [γ-$^{32}$P]ATP) containing 250 ng active Pak1 and 2.1 µg purified shootin1a as described (*Kubo et al., 2015*).

## In vitro binding assay

Purified GST-L1-CAM-ICD and shootin1a were incubated overnight at 4°C in reaction buffer (20 mM Tris-HCl pH 8.0, 100 mM NaCl, 1 mM EDTA, 1 mM DTT). After centrifugation for 15 min at 17,400 *g* at 4°C, the supernatants were incubated with Glutathione Sepharose 4B beads for 2 hr at 4°C. The beads were washed three times with wash buffer (20 mM Tris-HCl pH 8.0, 300 mM NaCl, 1 mM EDTA, 1 mM DTT) and once with TED buffer (20 mM Tris-HCl pH 8.0, 1 mM EDTA, 1 mM DTT). For elution, the Sepharose beads were incubated with 25 µl of elution buffer (15 mM reduced glutathione pH 8.0, 20 mM Tris-HCl pH 8.0, 100 mM NaCl, 1 mM EDTA, 1 mM DTT) for 2 hr at 4°C. For the binding assay in *Figure 3A*, we incubated 100 nM shootin1a and 100 nM GST-L1-CAM-ICD in 10 ml reaction buffer. After further incubation with Glutathione Sepharose 4B (bed volume 500 µl), GST-L1-CAM-ICD was eluted by 20 mM glutathione buffer (pH 8.0). After concentrating the 10 ml eluate with a centrifugal filter (Millipore), using half of the eluate, we could detect the interaction between shootin1a and L1-CAM-ICD by CBB staining. For the binding assay in *Figure 3B*, the supernatants were analyzed by immunoblot. Apparent dissociation constants were calculated by non-linear regression using GraphPad Prism 6 (GraphPad Prism Software).

## Immunoprecipitation and immunoblot

Immunoprecipitation and immunoblot were performed as described (*Toriyama et al., 2006*). For immunoprecipitation with HEK293T cells, cell lysates were prepared using NP40 lysis buffer (0.5% NP-40, 20 mM HEPES pH 7.5, 3 mM MgCl$_2$, 100 mM NaCl, 1 mM EGTA, 1 mM DTT, 1 mM PMSF, 0.01 mM leupeptin, 1 × PhosStop). The supernatants of cell lysates were incubated with 25 µl (bed volume) of anti-FLAG M2 gel (RRID:AB_10063035) overnight at 4°C. The anti-FLAG M2 gels were washed three times with wash buffer (0.1% Tween 20, 20 mM HEPES pH 7.5, 3 mM MgCl$_2$, 100 mM NaCl, 1 mM EGTA, 1 mM DTT) and once with TED buffer. To elute immunocomplexes, the gels were incubated with 25 µl of FLAG peptide (400 µg/ml) for 2 hr at 4°C. The immunocomplexes were analyzed by immunoblot.

For immunoprecipitation with cultured neurons, after netrin-1 (4.4 nM) stimulation for 1 hr, cell lysates were prepared with NP40-Triton lysis buffer (0.5% NP-40, 1% Triton X-100, 20 mM HEPES pH 7.5, 3 mM MgCl$_2$, 100 mM NaCl, 1 mM EGTA, 1 mM DTT, 1 mM PMSF, 0.01 mM leupeptin, 1 × PhosStop). The supernatants of the lysates were incubated with antibodies overnight at 4°C, and immunocomplexes were then precipitated with protein G-Sepharose 4B. After washing the beads with wash buffer (0.1% Tween 20, 20 mM HEPES pH 7.5, 3 mM MgCl$_2$, 100 mM NaCl, 1 mM EGTA, 1 mM DTT), immunocomplexes were analyzed by immunoblot.

## Immunocytochemistry and microscopy

Cultured neurons were fixed with 3.7% FA in Krebs buffer (118 mM NaCl, 5.7 mM KCl, 1.2 mM KH$_2$PO$_4$, 1.2 mM MgSO$_4$, 4.2 mM NaHCO$_3$, 2 mM CaCl$_2$, 10 mM Glucose, 400 mM Sucrose, 10 mM HEPES pH7.2) for 10 min at room temperature, followed by treatment for 15 min with 0.05% Triton X-100 in PBS on ice and 10% fetal bovine serum in PBS for 1 hr at room temperature. They were then labeled with antibodies, as described (*Shimada et al., 2008*). We used secondary antibodies conjugated with Alexa Fluor 488 or Alexa Fluor 594. For CMAC staining, cells were incubated with 2.5 µM CMAC for 2 hr before live-cell imaging. Fluorescence and phase-contrast images of neurons were acquired using a fluorescence microscope (Axioplan2; Carl Zeiss) equipped with a plan-

Neofluar 20 × 0.50 NA or 63x oil 1.40 NA objective (Carl Zeiss), a charge-coupled device camera (AxioCam MRm; Carl Zeiss), and imaging software (Axiovision3; Carl Zeiss). Live-cell images of cultured hippocampal neurons were acquired at 37°C using a fluorescence microscope (IX81; Olympus) equipped with an EM-CCD camera (Ixon DU888; Andor), using a plan-Fluar 20 × 0.45 NA or 40 × 0.60 NA objective (Olympus), and MetaMorph software. TIRF microscopy was performed using a TIRF microscope (IX81; Olympus) equipped with an EM-CCD camera (Ixon3; Andor), a CMOS camera (ORCA Flash4.0LT; Hamamatsu), a UAPON 100 × 1.49 NA (Olympus), and MetaMorph software. Axon length was measured using ImageJ (Fiji version).

## Fluorescent speckle imaging and traction force microscopy

The speckle imaging data in were obtained using neurons cultured on coverslips coated with L1-CAM-Fc or laminin as described (*Shimada et al., 2008*). Traction force microscopy was performed as described (*Toriyama et al., 2013*; *Abe et al., 2018*). Briefly, neurons were cultured on polyacrylamide gels with embedded fluorescent microspheres (200 nm diameter; Invitrogen). Time-lapse imaging of fluorescent beads and growth cones was performed at 37°C using a confocal microscope (LSM710; Carl Zeiss) equipped with a C-Apochromat 63x/1.2 W Corr objective. The growth cone area was determined by EGFP fluorescence or from DIC images. Traction forces under the growth cones were monitored by visualizing force-induced deformation of the elastic substrate, which is reflected by displacement of the beads from their original positions, and expressed as vectors. The force vectors detected by the beads under individual growth cones were then averaged, and were expressed as vectors composed of magnitude and angle ($\theta$) (*Figure 6—figure supplement 1A*, left panel) (*Toriyama et al., 2013*). To compare the forces under different conditions, the magnitude and angle ($\theta$) of the force vectors of the individual growth cones were statistically analyzed and expressed as means ± SEM, separately (*Abe et al., 2018*). They were also analyzed by one-way ANOVA with Tukey's post hoc test.

## Analyses of netrin-1 attached on the substrate

Netrin-1 attached on the substrate was analyzed as described (*Moore et al., 2012*) with modifications. Glass coverslips coated with polylysine or coated sequentially with polylysine and L1-CAM-Fc were incubated with culture medium and netrin-1 containing a 10-His tag at the C-terminus, in the absence or presence of 2 μg/ml heparin. The glasses were washed with PBS, and blocked for 1 hr at room temperature with 0.1% BSA in PBS. They were then labeled with anti-His antibody and secondary anti-mouse antibody conjugated with Alexa Fluor 488. Fluorescence images were acquired using a fluorescence microscope (IX81; Olympus) equipped with an EM-CCD camera (Ixon DU888; Andor), using a plan-Fluar 20 × 0.45 NA objective (Olympus), and quantified using ImageJ (Fiji version).

## Statistical analysis

All statistical analysis were performed using Microsoft Excel, Statistical software R (RRID:SCR_001905) and Graphpad prism 7 (RRID:SCR_002798). Significance was determined by the two-tailed unpaired Student's *t*-test in most cases. For multiple comparisons, we used one-way ANOVA with Schaffer's post hoc test or Tukey's post hoc test.

## Materials

Rabbit antiserum to shootin1a was raised by immunizing rabbits with the synthetic peptide CKGI-LASQ that corresponds to the region specific to shootin1a (*Higashiguchi et al., 2016*). The specificity of the antiserum was confirmed by immunoblot analysis (*Figure 1—figure supplement 3C*). Preparation and affinity purification of anti-pSer101-shootin1 and anti-pSer249-shootin1 antibodies are described elsewhere (*Toriyama et al., 2013*). Antibody against L1-CAM (Cat# sc-1508, RRID:AB_631086) was obtained from Santa Cruz. Antibodies against myc (Cat# 562–5, RRID:AB_591116) and FLAG (Cat# PM020, RRID:AB_591224) were obtained from MBL. Dulbecco's modified Eagle's medium, polylysine, Rabbit IgG control antibody (Cat# I8140, RRID:AB_1163661), anti-FLAG M2 antibody (Cat# F3165, RRID:AB_259529), anti-FLAG M2 gel (Cat# A2220, RRID:AB_10063035), FLAG peptide and heparin (Cat# H3149) were obtained from Sigma-Aldrich. Secondary anti-goat antibody conjugated with Alexa Fluor 488 (Cat# A-11055, RRID:AB_2534102), secondary anti-mouse antibody conjugated with Alexa Fluor 488 (Cat# A-11029, RRID:AB_2534088), active Pak1 and

CMAC were obtained from Invitrogen. Anti-His antibody (Cat# 014–23221) was obtained from Wako. Recombinant mouse netrin-1 containing a 10-His tag at the C-terminus was obtained from R and D Systems (Cat# 1109-N1-025, not CF form). Antibody against GST (Cat# 27-4577-01, RRID:AB_771432), Secondary anti-Rabbit antibody conjugated with HRP (Cat# NA934, RRID:AB_772206), Glutathione Sepharose 4B beads and PreScission protease were obtained from GE Healthcare. Secondary anti-Goat antibody conjugated with HRP (Cat# AP180P, RRID:AB_92573) and Amicon ultra-4 centrifugal filter devices were obtained from Millipore. Secondary anti-Rabbit antibody conjugated with Alexa Fluor 594 (Cat# 711-585-152, RRID:AB_2340621) was obtained from Jackson immune research. Fetal bovine serum and PhosStop were obtained from Japan Bio Serum and Roche, respectively.

## Acknowledgements

We thank Dr. Yuichi Sakumura (Nara Institute of Science and Technology) for critical reading of the manuscript, Ria Fajarwati Kastian (Nara Institute of Science and Technology) for technical support and Dr. Robert Farese, Jr. (Gladstone Institute, CA, USA), for providing the RF8 ES cells used to generate gene-targeting mice. This research was supported in part by a JSPS Grant-in-Aid for Scientific Research on Innovative Areas (JP25102010), JSPS KAKENHI (JP23370088 and JP26290007), AMED under Grant Number JP18gm0810011, the Osaka Medical Research Foundation for Incurable Diseases and the Takeda Science Foundation.

## Additional information

### Funding

| Funder | Grant reference number | Author |
|---|---|---|
| Osaka Medical Research Foundation for Intractable Diseases | | Kentarou Baba Michinori Toriyama Naoyuki Inagaki |
| Japan Society for the Promotion of Science | JP25102010 | Naoyuki Inagaki |
| Japan Agency for Medical Research and Development | JP17gm0810011 | Naoyuki Inagaki |
| Takeda Science Foundation | | Naoyuki Inagaki |
| Japan Society for the Promotion of Science | JP23370088 | Naoyuki Inagaki |
| Japan Society for the Promotion of Science | JP26290007 | Naoyuki Inagaki |

The funders had no role in study design, data collection and interpretation, or the decision to submit the work for publication.

### Author contributions

Kentarou Baba, Conceptualization, Data curation, Formal analysis, Validation, Investigation, Visualization, Methodology, Writing—original draft, Writing—review and editing; Wataru Yoshida, Data curation, Formal analysis, Investigation, Visualization, Methodology, Writing—review and editing; Michinori Toriyama, Data curation, Formal analysis, Validation, Investigation, Visualization, Methodology, Writing—original draft, Writing—review and editing; Tadayuki Shimada, Conceptualization, Data curation, Formal analysis, Validation, Investigation, Visualization, Methodology, Writing—review and editing; Colleen F Manning, James S Trimmer, Investigation, Methodology, Writing—review and editing; Michiko Saito, Resources, Investigation, Methodology; Kenji Kohno, Rikiya Watanabe, Resources, Investigation, Methodology, Writing—review and editing; Naoyuki Inagaki, Conceptualization, Resources, Supervision, Funding acquisition, Validation, Investigation, Visualization, Methodology, Writing—original draft, Project administration, Writing—review and editing

## Author ORCIDs

Tadayuki Shimada (iD) https://orcid.org/0000-0001-7333-9847
Michiko Saito (iD) https://orcid.org/0000-0002-6678-2135
Kenji Kohno (iD) http://orcid.org/0000-0002-3503-6551
James S Trimmer (iD) https://orcid.org/0000-0002-6117-3912
Naoyuki Inagaki (iD) http://orcid.org/0000-0002-2664-9196

## Ethics

Animal experimentation: This study was performed in strict accordance with the recommendations in the Guide for the Care and Use of Laboratory Animals of Nara Institute of Science and Technology. All relevant aspects of the experimental procedures were approved by the Institutional Animal Care and Use Committee of Nara Institute of Science and Technology (reference No. 1802).

## Decision letter and Author response

Decision letter https://doi.org/10.7554/eLife.34593.062
Author response https://doi.org/10.7554/eLife.34593.063

## Additional files

### Supplementary files

• Transparent reporting form
DOI: https://doi.org/10.7554/eLife.34593.059

### Data availability

All data generated or analysed during this study are included in the manuscript and supporting files. Source data files have been provided for Figures 1-8 and figure supplements.

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
