## [Decision Letter]

Thank you for submitting your article "Gradient-reading and mechano-effector machinery for netrin-1-induced axon guidance" for consideration by *eLife*. Your article has been reviewed by four peer reviewers, one of whom is a member of our Board of Reviewing Editors, and the evaluation has been overseen by Didier Stainier as the Senior Editor. The following individuals involved in review of your submission have agreed to reveal their identity: Simon Moore (Reviewer #2); Esther Stoeckli (Reviewer #4).

The reviewers have discussed the reviews with one another and the Reviewing Editor has drafted this decision to help you prepare a revised submission.

Summary:

The manuscript presents evidence that shootin1a phosphorylation, activated by netrin-1, promotes L1-CAM adhesive function, and that growth cones turn in response to a gradient of netrin-1 by differentially facilitating mechanotransduction by L1-CAM. The authors have previously reported that netrin-1 regulates the function of shootin1a as a clutch between cortactin and L1-CAM (Kubo et al., 2015). These findings are now extended by generating and characterizing a shootin1a knockout mouse line, identifying the shootin1a domain that binds L1-CAM, and addressing its role in actin retrograde flow and traction forces.

Essential revisions:

1) The reviewers raise several issues related to the in vitro axon turning assay. Netrin-1 binds tissue cultures surfaces and is argued to function as an immobilized cue. Does netrin-1 form a substrate adsorbed gradient in the assay employed? If so, is this functionally significant, and what are the characteristics of the gradient?

2) In the in vitro assay, disrupting shootin1a function compromises the length and rate of axon outgrowth. Can a compelling argument be made that turning is compromised and not just the capacity of the axon to extend?

3) Questions were raised related to why primary neurons from the shootin1a null mice were not tested in the turning assay, as opposed to RNA knockdown. Additional controls to increase confidence in the specificity of the RNA knockdown are suggested as well as testing shootin1a primary knockout neurons in order to make the results the assay more compelling.

4) The paper would be strengthened by clarification of the significance of the L1-CAM substrate for the turning response to netrin-1. Is the shootin1a mechanism limited to growth cones on an L1-CAM substrate, or is shootin1a more generally involved in turning responses to a netrin-1 gradient? This question is highlighted by finding no defects in commissural axon extension in the embryonic spinal cord, where a gradient of netrin-1 has been visualized. In general, the relationship between the phenotypes in vivo and the functions described in vitro were not clear (also see point 5 below).

5) The analyses of axon guidance phenotypes in vivo were considered rather weak and the conclusions drawn not compelling. The Shootin1a null mouse exhibits gross changes in brain development and architecture, strongly questioning the conclusion that the loss of major brain commissures is specifically due to axon guidance deficits. The reviewers' comments provide details and suggestions to improve the analysis of the shootin1a null phenotype.

The full reviews are included below for your reference, as they contain detailed and useful suggestions.

*Reviewer #1:*

The manuscript addresses the mechanisms employed by axonal growth cones to read a chemical gradient and the mechano-effector machinery that converts an extracellular environmental chemical signal into directional force. The authors present a technically impressive series of findings in support of a functional interaction between netrin-1 signaling and L1 adhesion in axon guidance. The paper argues that netrin-1 induces asymmetric phosphorylation of shootin1a across a growth cone to promote the mechano-effector function of L1, and that this is required to read a netrin-1 gradient. The findings are novel and the manuscript clearly written, well organized, and well illustrated. The following issues should be addressed before the manuscript is suitable for publication.

1) The findings presented in Figure 5E, Figure 6 and Figure 7C indicate that expression of the Shootin1a (1-125) mutant, or RNAi knockdown of shootin1a severely compromises the length and rate of axon outgrowth, in control conditions and in response to added netrin-1. The authors must address the relationship between axon outgrowth and turning. In other words, for an axon to turn, it must be able to grow. What compelling evidence indicates that turning is compromised and not just the capacity of the axon to extend?

2) The authors state "Our data demonstrate that a 0.4% (250:251) difference in netrin-1 concentration induces a 71% difference in shootin1a phosphorylation within growth cones, as a key readout of the spatial signal." A serious issue is the extent to which the authors can accurately infer the distribution of netrin-1 from the distribution of florescent BSA. BSA is a highly soluble protein. In contrast, netrin-1 rapidly binds tissue cultures surfaces with high affinity. A previous paper, Moore et al. (2012), reported that puffing netrin-1 from a pipette resulted in a gradient of netrin-1 attached to the cell-culture substrate, and that this patterned substrate of netrin-1 was sufficient to turn a growth cone. These and related findings have provided substantial evidence that netrin-1 does not function as a guidance cue while in solution, but instead that netrin-1 must be immobilized to a substrate for the growth cone to turn. Does netrin-1 form a substrate adsorbed gradient in the assay being employed by the authors? If so, is the difference across the axon still 0.4%? If a substrate bound netrin-1 gradient is present, is this what turns the axon? Determining if this is the case is critical as it will change the model (Figure 7E) from one in which only L1 functions as the force transducing entity, to a model where immobilized netrin-1 and L1 together transduce force and turn the growth cone.

3) Functional experiments are carried out using neurons with L1 presented as the only relevant substrate present. Can the authors clarify if the findings depend on the presence of L1? The authors provide evidence that netrin-1 induced phosphorylation of shootin1 promotes the interaction of shootin1 with L1, and this is required for netrin-1 induced axon guidance, but it is not clear if this guidance mechanism is specific to growth cones on an L1 substrate, or if it generalizes to non-L1 substrates. For example, netrin-1 is essential for commissural axons to extend to the ventral midline in the embryonic spinal cord. This is the one CNS region where a gradient of netrin-1 has been described, however it is also the one region described in the manuscript where the mechanism described clearly does not play a role. Loss of netrin-1, DCC or shootin1a result in similar defects in various brain commissures, but gradients of netrin-1 have not been described in these brain regions. It is critical for the authors to clarify how the results of the in vitro assays described map onto functional significance in vivo.

4) Related to #3 above, evidence is presented that dominant negative shootin1a (myc-shootin1a 1-125) increased the rate of retrograde F-actin flow in primary neurons cultured on an L1 substrate, consistent with shootin1a (1-125) inhibiting F-actin-adhesion coupling. Is this specific to the L1 substrate, or does the effect of disrupting shootin1a generally uncouple F-actin-adhesion coupling?

5) The requirement for Shootin1a in netrin-1 induced axon turning was assessed using miRNA knockdown. Can the authors clarify why primary neurons from the shootin1a null mice were not plated in the in vitro turning assay? Cells derived from a knockout avoid possible off target artefacts common with miRNA approaches.

Statistical analysis:

In Figure 2D, Figure 3—figure supplement 1, and Figure 4C, it appears that the variance has been removed from all of the control conditions. This is likely due to all control values all being first normalized to 1 and then averaged to a mean value of 1. It is invalid to arithmetically remove the variance from the control condition and then compare that population to the experimental condition with its variance intact. No valid conclusion can be drawn from this comparison.

The values of the control condition can be normalized to an average value of 1 while still maintaining the variance of the population. This can be done by obtaining the mean value for the controls, normalizing that to 1, and then applying the factor required to all of the raw data in both the control and experimental conditions. The controls will then have a mean value of 1 and a variance that can be compared to the mean value and variance of the re-scaled values for the experimental group. This will make the statistical comparison valid.

Figure 5D: The histograms presented lack error bars and no statistical analysis has been carried out.

*Reviewer #2:*

This manuscript further explores the role of the intracellular protein shootin1a in the gradient sensing and mechanical events that guide axons to netrin-1. Previously, the authors reported that netrin-1 regulates shootin1's clutch function between cortactin to L1-CAM (Kubo et al., 2015). In this report, the authors extend these findings by: (1) generating and characterizing a shootin1a knockout mouse line, (2) outlining the shootin1a domain that binds L1-CAM, and (3) further examining its role in actin retrograde flow and traction forces. The knockout is robustly confirmed by Southern, immunoblot and IHC analysis. Phenotypic consequences resemble that of netrin-1 knockout in certain contexts, but not in the spinal commissure. This discrepancy should be better addressed in the text given the extensive literature examining netrin-1 guidance of spinal commissural neurons (possible redundant proteins?). Using in vitro binding of purified proteins, as well as transfection of HEK293T and cortical neurons, previously reported interactions of shootin1 with cortactin (aa 261-377) are extended to show that the N-terminal region of (aa 1-125) associates with the intracellular domain of L1-CAM. They report that overexpression of shootin1a lacking the L1-CAM binding region disrupts netrin-1 induced reductions in actin retrograde flow and mechanical forces, supporting their proposed model that it bridges intracellular actin to extracellular L1-CAM anchor. Using RNAi and over expression of shootin1a lacking the L1-CAM binding region or a mutant that constitutively binds L1-CAM, the authors present evidence that shootin1a disruption slows axon extension and turning of primary hippocampal neurons toward a source of soluble netrin-1. A model is presented of an intracellular signaling pathway whereby extracellular soluble netrin-1, through DCC, activates Rac1 and Cdc42, leading to Pak1 mediated phosphorylation of shootin1 that then bridges it to retrogradely flowing actin through cortactin and to an extracellular 'adhesive substrate' through L1-CAM.

The major deficiency of this manuscript is how it overlooks netrin-1's diffusion characteristics and does not address that growth cone traction forces occur directly onto netrin-1. Specifically, their assumption that fluorescently tagged BSA mimics netrin-1 due to their similar size is over simplistic. Unlike soluble BSA, netrin-1 is detected in the membrane-bound fraction and not the soluble fraction (Serafini et al., 1994). Moreover, a study which I led, showed that fluorescently tagged netrin-1 efficiently binds to poly-lysine coated surfaces similar to the ones used in this study (Moore et al., 2012). The ability of traction forces to exerted directly on netrin-1, as well as the observation that only restrained netrin-1 turn axons should be addressed (Moore et al., 2009. Science 325:166). A simple, straightforward experiment that is within the technical repertoire of the authors would be to solubilize the netrin-1 by including 2ug/ml heparin in the media of their axon guidance assay. It may also be possible to localize netrin-1 in either the flow through of their chamber or attached to the cell culture surface.

*Reviewer #3:*

This is an interesting paper by Baba et al. that uses a wide variety of techniques to examine the role of Shootin1 downstream of netrin and L1 mediated axon guidance. The authors show that Shootin1 is expressed in many commissural axon pathways in the brain, but is not expressed by spinal commissural interneurons. They go on to generate a knock out mouse and show that Shootin1 KO mice have reduced axon commissures and some axon mis-routings. Using microfluidic devices, they show that a shallow gradient of netrin promotes a steep gradient of Phospho-shootin1 across hippocampal growth cones. Next using pull-down the authors show that Shootin1 binds the intracellular domain of L1-CAM, which is enhanced by phosphomimetic Shootin1-DD and phosphorylation by Pak1. Netrin also promotes phosphorylation of Shootin1 and Shootin1-L1 association. Shootin1 was previously shown by this group to bind Cortactin, but here they show that Shootin binds L1-CAM at its c-terminal, while binding Cortactin at its n-terminal. Interesting, they found that truncated Shootin1 1-125 could disrupt phosphomimetic Shootin1-DD binding to L1-CAM. Expressing Shootin1 1-125 in neurons increase F-actin retrograde flow (reduced clutching) and blocks increased clutching (reduced RF) promoted by Netrin, as well as inhibited the axon growth promoting effects of netrin. Finally, the authors show that Shootin1 knock-down or over expression of Shootin1 1-125 blocks turning toward netrin. The authors conclude that local Shootin1 phosphorylation by Pak1 downstream of netrin promotes axon guidance by increasing clutching of F-actin/Cortactin to L1-CAM in growth cones.

This is strong paper that presents novel and important findings and I find the data largely convincing. However, before publication I believe a few concerns must be considered. I detail my concerns by figure below.

First, why is there not even background labeling of Shootin1 in the spinal cord section (Figure 1—figure supplement 1C)? Second, from the knock out sections it is clear that there are dramatic defects in gross brain architecture, as the ventricles are extremely enlarged. This brings to question the specificity of the loss of commissures. Something should be mentioned about these gross defects and how long these mice survive. Clearly conditional KO will need to be performed someday. There is also oddly no background labeling in the KO section. While the authors may be commended for producing such "clean" data, readers may be concerned that identical imaging conditions were not used.

In Figure 2 the authors use AlexaFluor488-BSA as a proxy for Netrin. While it is true these molecules have similar MWs, they may bind the substratum differently. Netrin was shown to bind the substratum (PDL) (Moore et al., 2012), which was necessary for chemotropic axon guidance. It would be interesting to know if Netrin also binds to the PDL-L1 substrata used here. Would it be possible to either immunolabel bound netrin in microfluidic devices, or fluorescently label Netrin as was done for BSA? Since the authors make a point to argue that small difference in Netrin across the growth cone (.4% estimated from BSA gradient) are sufficient to promote turning, it is important to be certain this is really the concentration gradient of netrin. There is also no control for the phospho-Shootin1. I can think of several that would be worthwhile (DN-Pak, Shootin KD, function blocking Anti-DCC).

No issues with Figure 3 or 4.

In Figure 5 it would be interesting to see what Shootin1-DD expression does to RF rates and traction forces.

In Figure 6, there is very little axon extension in Shootin-RNAi neurons, which makes assessing turning difficult. Also, why did the authors not use neurons from Shootin KO animals? If using Shootin-RNAi for knock-down, the authors should show this construct is effective by blot and ICC?

The authors should consider mentioning important work by Nichol et al., 2016, which recently showed that axon guidance cues affect integrin receptor clutching forces to influence axon outgrowth and guidance. This work supports these earlier findings.

*Reviewer #4:*

Baba and colleagues describe an effect of Shootin1 as mediator of Netrin-induced growth cone guidance by linking L1CAM to the actin cytoskeleton. They have created a Shootin1 knockout mouse and show aberrant corpus callosum formation but not spinal cord commissure formation in the mutant. In the second part of the study, they use in vitro assays to demonstrate an interaction between Shootin1 and L1CAM in a phosphorylation-dependent manner mediated by Pak.

This is a potentially very interesting study, as it addresses the poorly understood linkage between surface receptors and the cytoskeleton. However, the study suffers from some quality issues that should be addressed before acceptance of the manuscript. The two parts of the study are not really linked and the in vivo part is rather preliminary and lacks quantitative assessment of the phenotype.

In general the in vivo part is rather weak, the claims are not supported by quantitative evidence. But above all, the phenotypes described in the first part do not seem to be related to the results from the second part, as the mutant mouse has major changes in brain development which preclude the analysis of Netrin/L1CAM interactions in response to Shootin1.

Figure 1, expression of Shootin1. Claims about the level of Shootin1 expression in mouse brains between E13.5 and P12 are made based on immunoblots without loading control.

The DAPI staining in Figure 1 looks very strange. I would expect DAPI throughout the brain but Figure 1A shows DAPI only in the edge of the tissue section but not within the tissue.

An effect on commissure formation in the Shootin1 KO mouse is not very surprising given that the entire brain morphology is severely affected (Figure 1C). Therefore, it is impossible to know, whether we are looking at comparable structures in Figure 1D-H.

The DiI tracings are not meaningful in a brain that looks like Figure 1C. The low resolution does not allow for any conclusion about axonal extensions or lack thereof.

Figure 1—figure supplement 1: Again there is an issue with the DAPI staining that only appears to be found in the edge of the tissue section not in the neurons within the brain. The TAG-1 staining is of poor quality. An effect of the absence of shootin1 on midline crossing of commissural axons in the spinal cord cannot be assessed at the low level of resolution. To me the commissure looks thinner in the mutant compared to the control section. A quantification of the phenotype is required.

Figure 2: The claim that phospho-Shootin1 is accumulated specifically on the growth cone side facing higher Netrin levels needs to be controlled for in a gradient of a control protein.

Figure 6: Cultures of hippocampal neurons were prepared on L1CAM substrate to measure the effect of Shootin1 and found an effect when downregulating Shooting with RNAi on growth and turning. Similarly, the perturbation of Shootin1-L1CAM interaction affected growth and turning but to a different degree. As growth rate was much less affected when only the interaction between Shootin1 and L1CAM was disturbed without lowering the Shootin1 levels. What about knockout neurons? I am surprised that the authors did not compare the effect of control neurons with neurons dissected from mutant animals. Or, why not using spinal cord neurons which do not seem to express Shootin1 (but do use L1CAM for growth) as a control.

How do the authors explain the difference seen in the rescue experiments. How can the DD version of Shootin1 rescue growth but not turning? To me this would call for an effect that is not L1CAM dependent. L1CAM is not required for turning in response to Netrin. Another control to address the issue of L1CAM/Netrin growth versus turning would be the use of an alternative substrate where growth cones migrate in an integrin-dependent manner. What happens then in response to Netrin?

---

## [Author Response]

Essential revisions:

*1) The reviewers raise several issues related to the* in vitro *axon turning assay. Netrin-1 binds tissue cultures surfaces and is argued to function as an immobilized cue. Does netrin-1 form a substrate adsorbed gradient in the assay employed? If so, is this functionally significant, and what are the characteristics of the gradient?*

Response 1: We appreciate these valuable comments of reviewers #1-#3. In response, we examined the attachment of netrin-1 on substrates coated with polylysine (conditions of Moore et al., 2012) and L1-CAM (our conditions). As reported by Moore et al. (2012), netrin-1 attached on the polylysine-coated substrate and was released by treatment with 2 μg/ml heparin (Figure 2―figure supplement 1A and B). Netrin-1 also attached to the substrate coated with L1-CAM. We also confirmed that our device produces a netrin-1 gradient attached to the L1-CAM–coated substrate in a manner dependent on the incubation time and that the difference across the growth cone is about 0.6-0.8% (Figure 2―figure supplement 1C), which is similar to that of BSA (Figure 2B). However, the amount of attached netrin-1 on L1-CAM was 39% of that on polylysine after 420 min incubation (Figure 2―figure supplement 1B), indicating that at least 61% of the applied netrin-1 is not attached to the substrate under our conditions.

Heparin treatment also removed netrin-1 from the L1-CAM–coated substrate (Figure 2—figure supplement 1A and B). In contrast to the data for spinal cord neurons (Mooreet al., 2012), the netrin-1 gradient induced axon turning of hippocampal neurons on the L1-CAM–coated substrate even in the presence of heparin (Figure 7—figure supplement 1). These results suggest that a gradient of soluble netrin-1 contributed to axon turning in the assay employed. However, the degree of netrin-1–induced axon outgrowth and turning was reduced in the presence of heparin (Figure 7—figure supplement 1C). These data are consistent with a previous report (Mai et al., 2009) that netrin-1 attached to the substrate induces axon turning of cultured hippocampal neurons. Thus, we conclude that the gradients of both the soluble and substrate-bound netrin-1 contribute to axon turning of hippocampal neurons in our assay system. In the revised manuscript, we have discussed these data in the modified text (subsection “Shallow gradients of netrin-1 elicit highly polarized shootin1a phosphorylation within growth cones”, second paragraph; subsection “Netrin-1–induced axon attraction requires shootin1a”, first paragraph).

As soluble and immobilized chemical cues can act as ligands to activate intracellular signaling pathways in growth cones (Huber et al., 2003; Kolodkin and Tessier-Lavigne, 2011; Lowery and Van Vactor, 2009), the present gradient-reading mechanism may explain both axonal chemotaxis and haptotaxis elicited by shallow gradients of soluble and substrate-bound netrin-1 (pink and yellow arrows, Figure 8E). We have discussed possible mechanisms for shootin1a-mediated axon guidance elicited by immobilized netrin-1 (subsection “Netrin-1–induced axon guidance through chemotaxis and haptotaxis”, first paragraph). We also referred to the model of Moore et al. (2012) where immobilized netrin-1 plays a key role in mechanical activation of FAK for axonal haptotaxis (see the aforementioned subsection, last paragraph).

2) In the in vitro assay, disrupting shootin1a function compromises the length and rate of axon outgrowth. Can a compelling argument be made that turning is compromised and not just the capacity of the axon to extend?

Response 2: Yes. As reviewers #1 and #3 pointed out, disrupting shootin1a function, by expression of the shootin1a (1-125) mutant or RNAi knockdown, compromised both axon outgrowth and turning. However, some of the neurons extended a relatively longer axon which did not turn (e.g., the yellow axon in Figure 7B and purple and pink axons in Figure 7F); we consider that the compromised turning of these axons cannot be explained by the inability of the axon to extend.

In our model, shootin1a regulates both axon outgrowth and axon turning: axon outgrowth is promoted by shootin1a-mediated clutch coupling (Shimada et al., 2008), while axon turning is regulated by the clutch coupling plus spatial regulation of the clutch coupling within the growth cone through netrin-1–induced shootin1a phosphorylation (Figure 8E). Displacement of wild-type shootin1a with constitutively active shootin1a (shootin1a-DD) can disturb netrin-1–induced shootin1a regulation without disturbing the clutch coupling, because shootin1a-DD can mediate clutch coupling (Toriyama et al., 2013) but cannot be regulated by phosphorylation. As demonstrated by the data in Figure 8B-D, disturbance of spatial shootin1a regulation within growth cones by shootin1a-DD compromised axon turning without inhibiting axon outgrowth. These data are consistent with our model (Figure 8E) and provide compelling evidence that shootin1a-mediated axon turning is compromised not just because the capacity of the axon to extend is inhibited. Thus, our answer to Reviewer #4’s question “How can the DD version of shootin1 rescue growth but not turning?” is “because shootin1a-DD can mediate clutch coupling but cannot be regulated by netrin-1 signals through phosphorylation”. We have explained this in the revised text (subsection “Netrin-1–induced axon attraction requires polarized shootin1a phosphorylation within growth cones”).

3) Questions were raised related to why primary neurons from the shootin1a null mice were not tested in the turning assay, as opposed to RNA knockdown. Additional controls to increase confidence in the specificity of the RNA knockdown are suggested as well as testing shootin1a primary knockout neurons in order to make the results the assay more compelling.

**Response 3:** Following the suggestions of the reviewers, to increase confidence in the specificity of the RNA knockdown, we performed the axon turning assay using primary neurons prepared from the shootin1a null mice. As shown in Figure 7―figure supplement 2, shootin1a knockout not only reduced the axon outgrowth velocity but also inhibited the growth cone turning toward the netrin-1 source. These data, together with the data in Figure 8 (see **response 2**), indicate that shootin1a plays an essential role in netrin-1–induced axon guidance.

*4) The paper would be strengthened by clarification of the significance of the L1-CAM substrate for the turning response to netrin-1. Is the shootin1a mechanism limited to growth cones on an L1-CAM substrate, or is shootin1a more generally involved in turning responses to a netrin-1 gradient? This question is highlighted by finding no defects in commissural axon extension in the embryonic spinal cord, where a gradient of netrin-1 has been visualized. In general, the relationship between the phenotypes* in vivo *and the functions described* in vitro *were not clear (also see point 5 below).*

Response 4: In response to the questions of reviewers #1 and #4, we performed an axon guidance assay on an alternative substrate, laminin. Laminins are widely used substrates for axon guidance assays (Turney and Bridgman, 2005; Nichol et al., 2016); L1-CAM on growth cones interacts directly with laminin presented on the substrate (Abe et al., 2018). In addition, integrin on growth cones also interacts with laminin, and laminin is thought to be involved integrin-dependent growth cone migration (Nichol et al., 2016). Growth cones of cultured hippocampal neurons on laminin turned in response to netrin-1 gradients (Figure 7—figure supplement 4A-C) as in the case of growth cones on L1-CAM. In response to the comments of reviewer #1, we also measured F-actin retrograde flow in growth cones on laminin. As we have recently reported (Abe et al., 2018), the F-actin retrograde flow rate in control growth cones on laminin was about 2.3 μm/min. As in the case of growth cones on L1-CAM (Figure 6A and B), overexpression of shootin1a (1-125) increased significantly the retrograde flow rate in growth cones on laminin (Figure 7—figure supplement 4D), indicating that shootin1a disruption also leads to F-actin-adhesion uncoupling in growth cones on laminin. Furthermore, uncoupling of F-actin-adhesion coupling by shootin1a (1-125) also inhibited netrin-1–induced axon turning on laminin (Figure 7—figure supplement 4A-C, 4E). Together, these data indicate that the shootin1a mechanism is not limited to growth cones on L1-CAM substrate and suggest that it is a more general mechanism.

As pointed out, we could not find defects in commissural axons of the embryonic spinal cord in shootin1 knockout mice. We have reconfirmed that there is almost no detectable shootin1a immunoreactivity in the ventral commissure of the spinal cord (please see Response 5a). Thus, no defects in extension of spinal commissural axons in shootin1 knockout mice can be explained by the absence of shootin1a in these axons, not by substrate specificity. These data suggest that the netrin-1–mediated guidance of the spinal commissural axons is regulated by a shootin1a-independent mechanism.

In response to the comment of reviewer #1, we have also clarified how the results of the in vitro assays correlate with functional significance in vivo, referring that netrin-1 gradients have not been reported at present in the forebrain commissure regions (subsection “Shootin1a–mediated axon guidance in the brain”).

5) The analyses of axon guidance phenotypes in vivo were considered rather weak and the conclusions drawn not compelling. The Shootin1a null mouse exhibits gross changes in brain development and architecture, strongly questioning the conclusion that the loss of major brain commissures is specifically due to axon guidance deficits. The reviewers' comments provide details and suggestions to improve the analysis of the shootin1a null phenotype.

Response 5: As pointed out, shootin1a knockout mice exhibit multiple defects in brain development and architecture. So, we agree that the dysgenesis of the corpus callosum, anterior commissure and hippocampal commissure cannot be attributed only to the axon outgrowth and guidance deficits observed byin vitro assays. However, we consider that dysgenesis of the commissural axons is at least consistent with the defects in axon outgrowth in vitro. In addition, the misprojection of the commissure axons in vivo is consistent with the defects in netrin-1–induced axon guidance in vitro. As reviewer #3 mentioned, generation of conditional knockout mice will be needed in future for detailed analyses of shootin1a functions in vivo. In the revised manuscript we discussed these points, mentioning that the dysgenesis of the forebrain commissures cannot be specifically attributed to axon guidance deficits (subsection “Shootin1a–mediated axon guidance in the brain”).

In the following, we respond to the comments of reviewers #3 and #4 regarding these points and in vivo analyses point-by-point:

Reviewer #3:First, why is there not even background labeling of Shootin1 in the spinal cord section (Figure 1—figure supplement 1C)? Second, from the knock out sections it is clear that there are dramatic defects in gross brain architecture, as the ventricles are extremely enlarged. This brings to question the specificity of the loss of commissures. Something should be mentioned about these gross defects and how long these mice survive. Clearly conditional KO will need to be performed someday. There is also oddly no background labeling in the KO section. While the authors may be commended for producing such "clean" data, readers may be concerned that identical imaging conditions were not used.

Response 5a to reviewer #3: In response to this comment, we performed immunohistochemistry of shootin1a in the cerebral cortex (P0; wild-type, heterozygous and homozygous) and spinal cord (E12.5; wild-type) under totally identical conditions [same thickness of the tissues (12 μm), same dilutions (x1000 for anti-shootin1a, x1000 for Alexa Fluor 594 anti-rabbit) and incubation conditions of the antibodies, same objective lens (x 20), same exposure time (133 msec), etc.] (please see Author response image 1). The sections were also counterstained with DAPI. All the images are raw data without manipulation. Prominent shootin1a signals were detected in the wild-type cortex, and the intensity of the shootin1a signals in the homozygous cortex was about half of that in the wild-type cortex (A). Under these conditions, we could not detect clear shootin1a (background) signals in the homozygous cortex (A) or wild-type spinal cord (B). Please note an intense nonspecific staining at the edge of the homozygous cortex (arrows, A), indicating that the sections were indeed incubated with the antibodies. Please also note the similar nonspecific staining at the edge of the wild-type spinal cord (Figure 1—figure supplement 2A). These data demonstrate that, with identical experimental and imaging conditions, wild-type cortices show prominent shootin1a signals while homozygous cortex and wild-type spinal cord exhibit almost no shootin1a signal. In other words, our data demonstrate that the background labeling of anti-shootin1a antibody is very low without manipulation.

As pointed out, shootin1 knockout mice exhibit multiple defects in the brain. We agree that the loss of commissures cannot be attributed only to the axon outgrowth and guidance deficits observed inin vitro assays. As mentioned, generation of conditional knockout mice will be required in future for detailed analyses of the functions of shootin1a in the brain. In the revised manuscript we have discussed these points (subsection “Shootin1a–mediated axon guidance in the brain”, last paragraph). Concerning the survival of shootin1 knockout mice, 13.3% of them (n = 98) died during the postnatal day 1-20. We have described these data in the revised text (subsection “Shootin1 knockout mice display abnormal projection of forebrain commissural axons”, first paragraph).

**Author response image 1. respfig1:** Shootin1a signals obtained under identical conditions.

Reviewer #4:In general the in vivo part is rather weak, the claims are not supported by quantitative evidence. But above all, the phenotypes described in the first part do not seem to be related to the results from the second part, as the mutant mouse has major changes in brain development which preclude the analysis of Netrin/L1CAM interactions in response to Shootin1.

Response 5b to reviewer #4: In response to this criticism, we quantified the thickness of the ventral spinal commissure using both TAG-1 staining and neurofilament staining; the thicknesses of the corpus callosum, anterior commissure and hippocampal commissure were quantified using Nissl staining. As shown in Figure 1—figure supplement 2B-C, the thickness of the ventral spinal commissure was not significantly different between wild-type and shootin1 knockout mice. On the other hand, the thicknesses of the corpus callosum, anterior commissure and hippocampal commissure were significantly reduced by shootin1 knockout (Figure 1—figure supplement 1D). We have included these data in the revised version.

As the reviewer pointed out, shootin1a knockout mice have major changes in brain development. We agree that the dysgenesis of the corpus callosum, anterior commissure and hippocampal commissure cannot be attributed only to the axon outgrowth and guidance deficits observed inin vitro assays. However, we consider that dysgenesis of the commissural axons is at least consistent with the defects in axon outgrowth in vitro. In addition, the misprojection of the commissural axons in vivo is consistent with the defects in netrin-1–induced axon guidance in vitro. As reviewer #3 mentioned, conditional knockout mice will need to be generated in future for detailed analyses of the shootin1 functions in vivo. In the revised manuscript, we have discussed these points (subsection “Shootin1a–mediated axon guidance in the brain”).

Figure 1, expression of Shootin1. Claims about the level of Shootin1 expression in mouse brains between E13.5 and P12 are made based on immunoblots without loading control.The DAPI staining in Figure 1 looks very strange. I would expect DAPI throughout the brain but Figure 1A shows DAPI only in the edge of the tissue section but not within the tissue.

Response 5c to reviewer #4: In the revised version, the immunoblot data of Figure 1—figure supplement 1A were replaced by those with loading controls (actin).In Figure 1 and Figure 1—figure supplement 1, DAPI was indeed detected within the tissue, but masked by the signal of shootin1a in the merged images. In the revised version, to demonstrate DAPI staining throughout the brain, we have provided the data for DAPI without shootin1a signals (Figure 1A and Figure 1—figure supplement 1B).

An effect on commissure formation in the Shootin1 KO mouse is not very surprising given that the entire brain morphology is severely affected (Figure 1C). Therefore, it is impossible to know, whether we are looking at comparable structures in Figure 1D-H.The DiI tracings are not meaningful in a brain that looks like Figure 1C. The low resolution does not allow for any conclusion about axonal extensions or lack thereof.Figure 1—figure supplement 1: Again there is an issue with the DAPI staining that only appears to be found in the edge of the tissue section not in the neurons within the brain. The TAG-1 staining is of poor quality. An effect of the absence of shootin1 on midline crossing of commissural axons in the spinal cord cannot be assessed at the low level of resolution. To me the commissure looks thinner in the mutant compared to the control section. A quantification of the phenotype is required.

Response 5d to reviewer #4: Concerning the resolution of the TAG-1 staining of commissural axons in the spinal cord, we have provided enlarged views of the commissural axons stained by anti-TAG-1 and anti-neurofilament antibodies (Figure 1—figure supplement 2B). Concerning the defects in commissure formation and the severely affected brain morphology, please see Response 5b. Concerning DAPI staining and quantitative analyses, see Responses 5c and 5b, respectively.

Response 6: In the following, we answer the comments of reviewers #1-#4 regarding citations and the statistical analyses point-by-point:

Reviewer #1:In Figure 2D, Figure 3—figure supplement 1, and Figure 4C, it appears that the variance has been removed from all of the control conditions. This is likely due to all control values all being first normalized to 1 and then averaged to a mean value of 1. It is invalid to arithmetically remove the variance from the control condition and then compare that population to the experimental condition with its variance intact. No valid conclusion can be drawn from this comparison.The values of the control condition can be normalized to an average value of 1 while still maintaining the variance of the population. This can be done by obtaining the mean value for the controls, normalizing that to 1, and then applying the factor required to all of the raw data in both the control and experimental conditions. The controls will then have a mean value of 1 and a variance that can be compared to the mean value and variance of the re-scaled values for the experimental group. This will make the statistical comparison valid.Figure 5D: The histograms presented lack error bars and no statistical analysis has been carried out.

Response 6a to reviewer #1: We thank the reviewer for these valuable suggestions. In the revised version, we have normalized values of the control condition to an average value of 1 while still maintaining the variance of the population (Figures 2D, 3E, 4B and 4D).

Concerning the traction force analysis (original Figure 5D): as described in the Materials and methods of the original version, we averaged the force vectors detected by the beads under individual growth cones, and expressed them as vectors composed of magnitude and angle (*θ*). To compare the forces under different conditions, the *force vectors* of individual growth cones on laminin or polylysine *were averaged*, and the magnitude and angle of the averaged force vector were obtained (Toriyama et al.,2013). In response to the reviewer’s comment, in the revised version, the *magnitude* and *angle (θ)* of the force vectors of the individual growth cones were statistically analyzed and *expressed as means ± SEM, separately.* They were also analyzed by one-way ANOVA with Tukey’s post hoc test. As shown in the revised Figure 6D, the magnitude of the force was increased significantly by netrin-1 stimulation. In addition, inhibition of the shootin1a–L1-CAM interaction by overexpression of myc-shootin1a (1-125) significantly decreased traction forces and abolished the netrin-1–induced increase in traction forces. We have described the new analysis method and these data in the revised text (subsection “Shootin1a–L1-CAM interaction mediates netrin-1–induced F-actin–adhesion coupling and mechanoresponse”, last paragraph and subsection “Fluorescent speckle imaging and traction force microscopy”).

Reviewer #3:The authors should consider mentioning important work by Nichol et al., 2016, which recently showed that axon guidance cues affect integrin receptor clutching forces to influence axon outgrowth and guidance. This work supports these earlier findings.

Response 6b to reviewer #3:We have mentioned the work by Nichol et al. (2016) in the revised version (subsection “Mechano-effector machinery for netrin-1–regulated axon guidance”, first paragraph).

Others

Reviewer #3:In Figure 5 it would be interesting to see what Shootin1-DD expression does to RF rates and traction forces.

Response 7a to reviewer #3:We previously reported the effects of shootin1a-DD expression on RF rates and traction forces (please see Figure 3D and E of Toriyama et al.,2013). Inhibition of Pak1 activity by dominant negative Pak1 (AID) increased RF rates and decreased traction forces. Co-expression of shootin1a-DD rescued the effects of AID, whereas co-expression of shootin1-WT did not rescue the effect of AID on RF rate or average traction stress. These data indicate that shootin1a-DD mediates clutch coupling and force generation in the absence of PAK1 activity. We have mentioned this in the revised text (subsection “Netrin-1–induced axon attraction requires polarized shootin1a phosphorylation within growth cones”).

There is also no control for the phospho-Shootin1. I can think of several that would be worthwhile (DN-Pak, Shootin KD, function blocking Anti-DCC).Reviewer #4:Figure 2: The claim that phospho-Shootin1 is accumulated specifically on the growth cone side facing higher Netrin levels needs to be controlled for in a gradient of a control protein.

Response 7b to reviewer #3 and #4: In response to these comments, we quantified phospho-shootin1a under the gradient of a control protein, BSA. As shown in Figure 2—figure supplement 2, the BSA gradient did not elicit polarized phosphorylation of shootin1a within growth cones.We have described these data in the revised version (subsection “Shallow gradients of netrin-1 elicit highly polarized shootin1a phosphorylation within growth cones”, last paragraph).